**Data availability statement:** All data files are available in public repositories. nsl kdd:

# Adaptive TreeHive: Ensemble of trees for enhancing imbalanced intrusion classification

**Mahbub E. Sobhani**[1], **Anika Tasnim Rodela**[1], **Dewan Md. Farid**[2]*

**1** Department of Computer Science and Engineering, United International University, United City, Dhaka, Bangladesh, **2** Department of Computer Science and Engineering, Southeast University, Tejgaon Industrial Area, Dhaka, Bangladesh

* dewanfarid@seu.edu.bd

## Abstract

Imbalanced intrusion classification is a complex and challenging task as there are few number of instances/intrusions generally considered as minority instances/intrusions in the imbalanced intrusion datasets. Data sampling methods such as over-sampling and under-sampling methods are commonly applied for dealing with imbalanced intrusion data. In over-sampling, synthetic minority instances are generated e.g. SMOTE (Synthetic Minority Over-sampling Technique) and on the contrary, under-sampling methods remove the majority-class instances to create balanced data e.g. random under-sampling. Both over-sampling and under-sampling methods have the disadvantages as over-sampling technique creates overfitting and under-sampling technique ignores a large portion of the data. Ensemble learning in supervised machine learning is also a common technique for handling imbalanced data. Random Forest and Bagging techniques address the overfitting problem, and Boosting (AdaBoost) gives more attention to the minority-class instances in its iterations. In this paper, we have proposed a method for selecting the most informative instances that represent the overall dataset. We have applied both over-sampling and under-sampling techniques to balance the data by employing the majority and minority informative instances. We have used Random Forest, Bagging, and Boosting (AdaBoost) algorithms and have compared their performances. We have used decision tree (C4.5) as the base classifier of Random Forest and AdaBoost classifiers and naïve Bayes classifier as the base classifier of the Bagging model. The proposed method Adaptive TreeHive addresses both the issues of imbalanced ratio and high dimensionality, resulting in reduced computational power and execution time requirements. We have evaluated the proposed Adaptive TreeHive method using five large-scale public benchmark datasets. The experimental results, compared to data balancing methods such as under-sampling and over-sampling, exhibit superior performance of the Adaptive TreeHive with accuracy rates of 99.96%, 85.65%, 99.83%, 99.77%, and 95.54% on the NSL-KDD, UNSW-NB15, CIC-IDS2017, CSE-CIC-IDS2018, and CICDDoS2019 datasets, respectively, establishing the Adaptive TreeHive as a superior performer compared to the traditional ensemble classifiers.

https://www.kaggle.com/datasets/hassan06/nslkddunsw-nb15: https://www.kaggle.com/datasets/dhoogla/unswnb15cic-ids2017: https://www.kaggle.com/datasets/dhoogla/cicids2017cic-ids2018: https://www.kaggle.com/datasets/dhoogla/csecicids2018. CIC-DDoS2019: https://www.kaggle.com/datasets/dhoogla/cicddos2019.

**Funding:** The author(s) received no specific funding for this work.

**Competing interests:** No authors have competing interests.

## Introduction

Intrusion classification holds paramount worth in ensuring the utmost clarity and vigilance in cybersecurity. Computer networks have become the backbone of important activities such as communication, commerce, data storage, and industrial automation in today's interconnected and technologically driven society. Despite the various technological advancements we've achieved, the area of cyberspace remains susceptible to a plethora of risks. The possibility of cyber-attacks and unauthorized access is omnipresent and can be exemplified at any given moment [1]. Intrusion classification is critical in protecting information systems' integrity, confidentiality, and availability by enabling administrators to detect and respond to cyber threats in real time. Intrusion classification is the cybernetic alert mechanism deployed within a computational ecosystem, designed to distinguish, categorize, and respond to malicious patterns or deviations from conventional behavioral norms, thereby fortifying the system's robustness against unauthorized access or adversarial activities where the main objective is to detect and respond to unauthorized or unusual activity on a network or host [2]. Intrusion Detection Systems (IDSs) depicted in Fig 1 stretch back to the late 1980s [3], primarily focusing on host-based [4] detection before evolving into network-based [5] systems. There are two types of IDS [6]: Network-Based IDS (NIDS) [7] and Host-Based IDS (HIDS). NIDS examines data packets at the network perimeter and notifies administrators if malicious traffic is detected. HIDS [7] are deployed on separate hosts or devices to monitor local activities and trigger alarms for unusual behavior that may go undetected at the network level. As the internet expanded and new attack categories emerged, it became challenging to intrusions accurately and quickly. Therefore, it is crucial to use big data techniques [8] to identify unauthorized access efficiently. Thus, the imperative is to design an intrusion classification model tailored to the distinctive characteristics of big data, specifically preaching the challenge of securing communication networks. Although there is a rising popularity of utilizing machine-learning based methods to classify malware, there is a significant lack of focus on accurately classifying intrusions in intelligent models [9] that are capable of effectively safeguarding communication networks against modern attacks. This neglect highlights the vulnerability of communication networks to evolving threats and emphasizes the importance of creating machine-learning models specifically designed to counter contemporary attacks. The primary hindrances in the advancement of classifying intrusions revolve around the utilization of machine learning-based methods for categorizing both modern and outdated [10] malware-challenges that we have discovered remained unresolved in earlier efforts. Furthermore, the scarcity of a publicly available large-scale balanced dataset for intrusion classification poses another inevitable constraint in the goal of developing quite precise models.

In the past decade, numerous approaches have been found in intrusion classification and detection to classify malware accurately and quickly. A wide range of machine learning (ML) and deep learning (DL) algorithms have been used to create comprehensive intrusion classifiers. ML models can classify and learn new patterns from unlabeled data without explicit programming [11]. Moreover, commonly used algorithms for classification tasks include Random Forest (RF) [12], Support Vector Machine (SVM) [13], Decision Tree (DT) [14,15], Adaptive Boosting (AdaBoost) [16], Extreme Gradient Boosting (XGBoost) [17] has notably influenced the cybersecurity domain, eventually enhancing its overall effectiveness. In contrast, DL models such as Convolutional Neural Network (CNN) [18], and Long Short-Term Memory (LSTM) [19] portray noteworthy results on intrusion detection datasets by successfully handling the labeled data [20]. However, it is important to act with caution and avoid applying deep learning algorithms when they are not critical. The impact of tree-based methods such as DT [21] and RF [22] has been emphasized in earlier research. These approaches

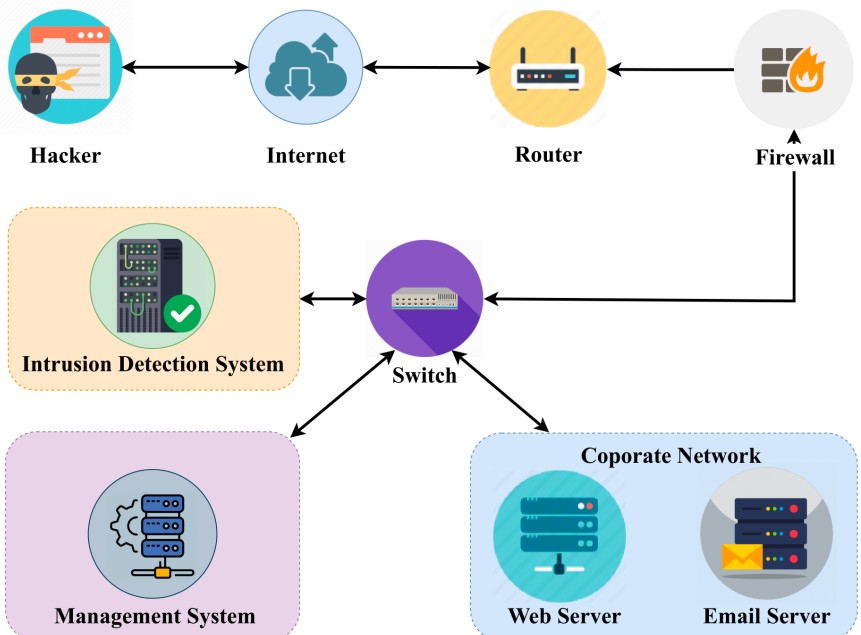

**Fig 1. Intrusion detection systems and local area networks.** The diagram illustrates a corporate network protected by a layered security architecture. External threats originating from the internet or a hacker are routed through a firewall and switch before reaching the internal network components, which include a web server, email server, and a centralized management system. An Intrusion Detection System (IDS) monitors the traffic passing through the network to detect and report suspicious activities. The design emphasizes the role of the IDS in safeguarding network assets by identifying potential security breaches in real-time.

have proven to be successful in handling modern and outdated attacks during intrusion classification. Harmonizing unsupervised and supervised methods synergistically induces good performance, exemplified by utilizing bagging with clustering techniques [23]. Nevertheless, the experimental setup lacked a large-scale experiment. The constraints of rule-based [22,24] methods in the intrusion classification task include scarcity of flexibility, coverage limitation, challenges in rule optimization, difficulty in handling attack variation, incapability in handling ambiguity, and little classification capabilities, which have provoked researchers to scrutinize big-data-driven [25] techniques as a means to overcome these constraints and improve detection rate. In recent years, transfer learning-based approaches [26] have demonstrated promising accomplishments in the intrusion classification task due to their intrinsic capability to capture feature patterns and decipher interrelated dependencies. Lately, artificial neural network-based methods [27,28] depicted good prowess in several intrusion classification datasets, including KDD-CUP99, NSL-KDD, and CSE-CIC-IDS2018, to name a few. Among LSTM-based methods, Mahdavisharif et al. [29] detected the malicious attacks presented in the NSL-KDD dataset. However, several contemporary attacks need to be addressed in the experiment. Moreover, it should be pointed out that DL models may not be competent in covering a considerable proportion of attacks where the dataset is comparatively small [30]. Amid the ensemble-based approach, Zoppi et al. [31] utilized an ensemble of unsupervised algorithms, mentioning promising results, but their proposed model suffered from a generalization capability on different datasets. However, feature selection-centric approaches have started exhibiting outstanding performance, where Rodriguez et al. [32] increased the computational efficiency considering performance loss. Likewise, Zhang et al. [33] took a similar

approach, combining with ensemble methods. In contrast, Stiawan et al. [34] preached the essence of relevant features and utilized different subsets of features as training data, which resulted in comparatively good performance despite longer execution time. Surprisingly, the harmony of supervised and unsupervised methods has yet to be applied in any study for intrusion classification tasks in large-scale experiments with different up-to-date benchmark datasets. Henceforth, we endeavor to utilize the formidable strengths of clustering, incorporating rule-based approaches as we delve into an exploration of their unexplored capabilities in the domain of intrusion classification.

In this paper, we have addressed several constraints associated with intrusion classification, especially concerning the lack of a futuristic approach that integrates extracting informative instances in an unsupervised manner and classification through tree-based methods. The objective is to overcome the identified limitations and come up with a concrete foundation for additional improvements in the field of intrusion classification while mitigating the challenge of large-scale balanced datasets [35,36], thereby paving the way for future research work. To achieve this, multiple comprehensive balanced datasets have been curated by rigorously harnessing clustering and non-parametric supervised algorithms. Moreover, a tree-based collaborative prediction process named Adaptive TreeHive has been proposed for the intrusion classification task, wherein we have optimized the complexity and computational efficiency through a uniquely tailored architecture consisting of multiple decision trees incorporating data randomization and dimensionality reduction. Additionally, scrutiny has been performed to confirm whether the performance of Adaptive TreeHive is improved by harnessing the harmony of tree-based collaborative prediction-making. In short, the proposed Adaptive TreeHive accepts subsets of informative instances as input, which is subsequently standardized for achieving better convergence. The instances are then fed into the decision nodes, which undergo a sequence of rules to classify intrusions. The contribution of this study is summarized below:

1. We have developed five large-scale balanced datasets for the intrusion classification task, which have been carefully crafted by utilizing clustering and a non-parametric supervised algorithm e.g. SMOTE.
2. The robustness of our curated datasets has been meticulously scrutinized by employing over-sampling and under-sampling techniques on the imbalanced datasets.
3. A state-of-the-art random tree-based collaborative prediction-enabled tailored architecture named Adaptive TreeHive has been introduced, demonstrating enhancements in the intrusion classification task compared to existing ensemble learning methods.
4. The effect of the training data size on the effectiveness of the proposed method Adaptive TreeHive in correcting detection errors in intrusion classification has been investigated.
5. The results of the proposed Adaptive TreeHive have been juxtaposed with several baseline intrusion classification models, demonstrating its superiority in the task by outperforming previous state-of-the-art ensemble models with reduced dimension and quick classification. This showcases its consistent performance across five large-scale public benchmark datasets, providing evidence of its excellent generalization capability across diverse types of attack categories.

## Background study

The digital revolution brings both advanced tech capabilities and heightened cyber threats, making intrusion classification pivotal in modern cybersecurity. Evolving from signature-based to ML-driven anomaly detection, intrusion classification now requires a

comprehensive understanding to build effective security strategies. The task of intrusion classification has gained noteworthy attention, leading to the emergence of novel insights in methods and datasets. The study of data balancing has indeed garnered steeped attention, it is evident that significant standards have not yet been met. This literature review provides an in-depth assessment of intrusion classification techniques for high and low-dimensional datasets, ranging from traditional to trendy methods while highlighting the methodologies, guiding principles, benefits, and limitations.

Injadat et al. (2020) [22] developed a ML-based IDS framework that had been optimized in multiple stages. Their purpose was to build a model that would be computationally less costly while maintaining high detection performance. They used the Z-score method for data normalization and SMOTE was used for minority class oversampling. They established the smallest advisable training sample size after discovering that SMOTE can minimize the training sample size. They used CIC-IDS2017 and UNSW-NB15 benchmark datasets for their model training and evaluation. Additionally, they compared two feature selection techniques which are called Information Gain-based Feature Selection (IGBFS) and Correlation-based Feature Selection (CBFS). Experimental hyperparameter tuning on K-nearest neighbors (KNN) [37] and RF classifiers was also a part of their research. According to the findings of their study, an optimized RF classifier with Bayesian Optimization using Tree Parzen Estimator (BO-TPE-RF) with the IGBFS method was able to classify targets with 99% accuracy on both datasets and reduced the false alarm rate by 1-2%. Zoppi et al. (2019) [31] discussed their investigation of meta-learning approaches that rely on ensembles of unsupervised algorithms. Observing different meta-learning approaches through ensembles of unsupervised base learners [38] helped them explore that specific meta-learning approaches significantly reduce misclassification compared to non-meta unsupervised algorithms. They used 21 datasets, 9 different meta-learning approaches, and 15 unsupervised algorithms for this experiment. They focused on network attacks and bio-metric authentication processes. In search of robust meta-learners, they also discussed the impact of base-learners that rely on multiple algorithms, such as stacking, cascading, delegating, (weighted) voting, and cascade generalization. They used different evaluation metrics, but primarily focused on the Matthews Coefficient (MCC). Parameter tuning and the heterogeneity of public datasets are two of the biggest limitations to building a robust network IDS. They concluded that a combination of both supervised and unsupervised algorithms is recommended for optimal results, and it was not possible to find an unsupervised algorithm that will outperform all IDS datasets. Ogobuchi et al. (2022) [21] proposed BoostedEnML as an ensemble model created using the best-performing boosting classifiers from their experiment. DT, RF, ET, LGBM, AD, and XGB were used to obtain an ensemble using the stacking method and majority voting approach. Two ensemble models based on boosting techniques (XGB and LGBM) were used to suggest an ensemble model using the stacking methodology. They solved the data imbalanced problem of CICIDS2017 and CICIDS2018 by using the SMOTE technique (SMOTE) and adaptive synthetic (ADASYN) [39] techniques. They concatenated all the CSV files into a single file for both datasets to obtain a robust dataset. K-fold was used to split the data into training, validation, and testing sets. During this research, they also performed statistical analysis including uni-variate, bi-variate, and multivariate using different data visualization tools. They used MaxAbsScaler in the preprocessing of the dataset. They used different evaluation metrics such as accuracy, precision, recall, f-score, and AUC to justify the robustness of their model. They achieved almost 100% accuracy in each of the datasets for multi-class classification.

Ogobuchi et al. (2022) [26] introduced ELETL-IDS (Efficient-Lightweight Ensemble Transfer Learning), an ensemble model designed using the model averaging approach. They proposed a transfer learning IDS based on a CNN architecture. The best three performing models

(InceptionV3, MobileNetV3Small, and EfficientNetV2B0) obtained from their experiment were selected to develop their model. They concatenated all the CSV files into a single file for each dataset (CICIDS2017, CSECICIDS2018) to obtain a robust dataset. They selected the features for model training by using the Random Forest Feature Importance (RFFI) provided by the Sklearn Library. SMOTE and Borderline_Smote were used to solve the data imbalance problem. They followed Quantile Transformation for numericalization in data preprocessing. They prepared the data in a suitable input format for using the pre-trained models, allowing CNN to learn all the patterns easily. BO-TPE was used for fine-tuning, and they achieved a shallow false positive rate with 100% accuracy. Zhang et al. (2022) [33] presented an effective ensemble-based automatic feature selection method (EAFS) for intrusion detection to overcome computationally complex and time-consuming feature selection methods. The authors proposed a novel approach that greatly improves the accuracy and efficiency of the detection system. They used UNSW-NB15, CICIDS2017, and CSE-CICIDS2018 benchmark datasets for their experiment. They removed features with zero variance using a variance threshold and the importance of each feature was obtained and added to the selected subsets. The effectiveness of the subsets was evaluated with the Normalized Subset Objective Measure (NSOM) scores. As the final selection, the subset with the highest NSOM value was selected. NSOM is a metric established for evaluating the efficacy of a subset of network intrusion detection features. The accuracy of the subset, the number of features in the subset, and the duration of training were considered by NSOM. When the authors compared their method to previous research on this dataset, the results showed that their method was more efficient in identifying the most beneficial data for intrusion detection. Mahdavisharif et al. (2021) [29] proposed a method for deep learning based on LSTM to detect intrusions in communication networks. This model could recognize complicated relationships as well as long-term interdependence between incoming traffic packets. BigDL, a distributed deep-learning framework for Apache Spark, was used to train the model on the NSL-KDD dataset, and the proposed method was named BDL-IDS. The suggested system, BDL, combined LSTM blocks to boost network efficiency. It could recall past experiences and identify long-term patterns. It had a total of three layers: an input layer, an output layer, and a hidden layer made up of LSTM cell blocks. Each of the 41 input neurons in the NSL-KDD had been connected to a Memory Cell Block in the hidden layer. Big data and deep learning may assist IDS to become more precise and faster, while additionally decreasing false alarms. Distributed and parallel computation can speed up the system.

FatimaEzzahra et al. (2020) [40] mentioned and compared three multi-layer LSTM-based intrusion detection models. IDS, according to the authors, relied on shallow learning and individual feature engineering, which could be inadequate for dealing with enormous quantities of data and real-time environmental constraints. Deep learning models, such as LSTM [41], can handle vast amounts of data without the need for manual feature engineering [42]. The authors used PCA and Mutual Information as reducing dimensionality and feature selection techniques to create LSTM-based IDSs without implementing any dimensionality reduction techniques. They tested their approach on a benchmark dataset, KDD99, and found that models based on PCA achieved the best accuracy for training and testing in both binary (99.44%) and multi-class (99.39%) classification. The authors provided valuable insights into how deep learning solutions like LSTM can improve the accuracy of IDSs by handling large amounts of data without requiring manual feature engineering. Talukder et al. (2023) [43] proposed an in-depth investigation into the construction of a reliable hybrid machine-learning model for network intrusion detection. To increase detection rates and reliability, an approach involving ML and deep learning methods is proposed. To achieve effective pre-processing, the authors used SMOTE for data balance and XGBoost for choosing features.

Furthermore, a deep learning-based feature selection is used to decrease dimensionality, discard redundant features, filter unneeded data, and simplify the process while boosting detection abilities. They split the processed dataset into training and testing sets using K-fold cross-validation. They evaluated different ML and DL models to find the best classification model for both binary and multi-class classification. With the chosen features, RF achieved the highest accuracy rate on KDDCUP'99 (99.99%) and CIC-MalMem-2022 (100%) datasets. Kasongo et al. (2020) [27] presented an extensive analysis of the IDS and proposed a feature selection-based approach to address the issue of false positive rate and low detection accuracy. They discussed various ML techniques used in IDS research. The UNSW-NB15 dataset was used for this research and a detailed analysis of this dataset was provided. A filter-based feature selection technique was applied using the XGBoost algorithm and the vector space was then fed to various classification algorithms such as SVM, k-Nearest-Neighbour (KNN), Logistic Regression (LR) [44], Artificial Neural Network (ANN), and DT. Both binary and multi-class classifications were considered. A reduced feature vector containing 19 features was taken for binary and multi-class classification. The DT achieved the best result with 90.85% accuracy on binary classification, while ANN achieved the best result with 77.51% accuracy on multi-class classification.

Karatas et al. (2020) [45] proposed six ML algorithms to build a more realistic IDS using an up-to-date security dataset called CSE-CIC-IDS2018. The selected dataset was also imbalanced, but they reduced the imbalance ratio using the SMOTE Technique (SMOTE), and the minority class numbers were increased with the help of this technique. The main aim of this research was to detect rarely encountered attacks accurately, as IDSs are generally trained and evaluated using pre-collected datasets. They evaluated their proposed system using Accuracy, Precision, Recall, F1-Score, and Error Rate values. They used a data sampling model to generate new data for the minority class. On accuracy measurement of the system, they found the best result using AdaBoost on both the original and sampled datasets. Sethi et al. (2021) [46] suggested a unique approach that employs a reinforcement learning-based IDS with Deep Q-Network logic applied to several distributed agents for more accurate detection and classification. They also used attention mechanisms. In this approach, agents work together to provide a standard security system. Deep Q-Network logic was implemented on multiple distributed agents in this IDS system for accurate malicious attack detection. Deep Q-Network is a variant of reinforcement learning. Denoising autoencoder (DAE) [47] is a concept that helps preprocess input data by decreasing noise and selecting the most relevant features, which helps in the elimination of biases from the model. They ran comprehensive studies with and without DAE on the NSL-KDD and CIC-IDS2017 benchmark datasets and compared the results of the studies. Their experiment indicates that they outperformed the state-of-the-art works for CIC-IDS2017 and produced good results on NSL-KDD. Zhendong et al. (2020) [48] addressed the issue of traditional ML methods' high false alarm rate and proposed an innovative intrusion detection approach called Semantic Re-encoding and Deep Learning (SRDLM). This method re-encoded network traffic, leading to a new representation of the data and improving the model's differentiating capability to detect malicious and non-malicious attacks. The above approach also improved the IDS's generalization capability. They used the NSL-KDD benchmark dataset for their experiment and discovered the data using PCA. In this study, they used ResNet_8, ResNet_20, and ResNet_56 for intrusion detection. The 20-layer and 56-layer ResNet [49] did not affect any significant improvement in detection, so they chose ResNet_8 for the following experiment. ResNet with semantic re-encoding increased the detection capability to 94.03% accuracy.

Nadir et al. (2023) [28] proposed a novel approach for constructing an efficient intrusion detection and prevention system for computer servers. They used the KDD-CUP99 benchmark dataset for their experiment. After pre-processing and feature extraction, they used Firefly Optimization to prepare the data for the training and testing phases. FFO is a population-based meta-heuristic algorithm that follows a stochastic optimization technique, which can help to select the most informative features that can give the model better generalization and detection capability. To improve attack recognition, they used min-max normalization in the pre-processing phase. Then, a Probabilistic Neural Network (PNN) was used for better pattern identification and classification. The proposed model, Firefly Optimization and Probabilistic Neural Network (FFO-PNN) achieved 98.99% accuracy with reduced training time and better performance measures. Stiawan et al. (2020) [34] addressed the importance of selecting significant features as a subset of the original dataset for better performance of an IDS model. Information Gain was used as a feature selection technique in this research. According to the minimum score values, the features were distributed into groups. 20% of the CIC-IDS2017 dataset was taken and split into 70% for training and 30% for testing data. After selecting the most significant features, different ML techniques such as RF, Bayes Net (BN), Random Tree (RT), Naive Bayes, and J48 classifiers were used. TPR, FPR, Precision, Recall, Accuracy, percentage of incorrectly classified, and execution time were used for analysis to evaluate the model. The experimental results showed that RF with Information Gain achieved the best result with 99.86% accuracy using 22 relevant features, while J48 achieved 99.87% accuracy using 52 features with a lengthy execution time. Rodriguez et al. (2022) [32] addressed the issue of traditional ML models' inability to detect new attacks, rather than known attacks. The authors analyzed different ML techniques for binary and multi-class classification, which had more detection capability for new attacks. They used the CIC-IDS2017 dataset for their experiment, as it contained different types of up-to-date attacks. Their experiment results showed that reducing redundant features using correlation-based feature selection (CFS) helped ML models require less time with only a slight performance loss. They found that tree-based machine-learning techniques showed better attack detection than complex algorithms. The classification score obtained F1 values of over 0.999 in the full dataset, 0.990 with the CFS-based attribute selection method, and 0.997 using Zeek-derived flows and attributes.

## Machine learning algorithms

In the context of anomaly classification, the initial step involves specifying a baseline for diverse network behaviors. This entails harnessing machine learning algorithms to train and learn the patterns and behaviors associated with both anomalies and normal attacks. The literature offers a diverse selection of machine-learning algorithms. To choose the most suitable one for our specific needs, we have diligently implemented five of them, which are detailed in the ensuing section.

### Unsupervised learning

A commonly used approach called clustering looks to identify clusters of related data based on similarities in features. This method has no set goals, so there is no need to tell the algorithm how to arrange things because clusters appear naturally. As a result, observations within the same cluster are more similar than those in other clusters. The main goal is to improve the similarity within clusters while maximizing the dissimilarity between groups/clusters. K-Means, which is known for its versatility, is a useful tool for exploratory analysis as it effectively handles a variety of data types, including images, figures, and text. Clustering,

particularly K-Means, is an efficient method for data categorization in the field of unsupervised learning. Unlike its supervised counterpart, unsupervised learning involves algorithmic exploration of data patterns without the help of output variables. Rather than using predetermined outcome metrics, it works based on inherent features. In a formal sense, the goal is to ascertain (1):

$$\arg\min_{\mathbf{S}} \sum_{i=1}^{k} \sum_{\mathbf{x} \in S_i} \|\mathbf{x} - \mu_i\|^2 = \arg\min_{\mathbf{S}} \sum_{i=1}^{k} |S_i| \operatorname{Var} S_i \tag{1}$$

where $\mu_i$ (2) is the mean (centroid) of points in $S_i$ and $|S_i|$ denotes the size of $S_i$ and $\|.\|$ is the usual $\|L\|^2$ (3) norm. As an outcome, the pairwise squared deviations of points within the same cluster are reduced (4)

$$\mu_i = \left|\frac{1}{S_i}\right| \sum_{x \in S_i} x \tag{2}$$

$$|x| = \sqrt{\sum_{k=1}^{n} |x_k|^2} \tag{3}$$

$$\arg\min_{\mathbf{S}} \sum_{i=1}^{k} \frac{1}{|S_i|} \sum_{x,y \in S_i} \|x - y\|^2 \tag{4}$$

K-Means is one of the most popular and widely used approaches in the range of clustering techniques. K-means divides data into clusters according to the user-defined value, K, using an iterative refinement process. Iteratively shifting cluster centers to converge on the ultimate clustering configuration requires initializing cluster centers using random data points. The value of K, which stands for the required number of centroids, is essential for how the algorithm works. The initial formation of centroids, which act as cluster centers, is random. K-Means reduces Euclidean distances (5) by selecting the centroid closest to each data point.

$$d(x, c) = \sqrt{\sum_{i=1}^{n} (x_i - c_i)^2} \tag{5}$$

With $n$ is the number of dimensions (features) in the data points. $x_i$ is the $i$th feature value of data point $x$. $c_i$ is the $i$th feature value of cluster centroid $c$. Centroids are then updated by calculating the mean of the data points within a cluster, thus reducing intra-cluster variance. The algorithm continues to iterate between the assignment and recalculation phases until the convergence requirements are met. It is common practice to run the K-Means clustering algorithm multiple times with different starting points, as the initial result may not be the most effective. To evaluate various outcomes, multiple initiating techniques such as the Forgy and Kaufman techniques are used. There is no exact correct strategy for determining the appropriate value of K, or the number of centroids to be generated. A popular method for determining the ideal number of clusters is to measure the sum of squared errors for various K values and identify the point, commonly referred to as the "elbow point," [50] that offers the lowest error sum. The best number of clusters for the algorithm can be determined using this point. There are several limitations associated with K-Means clustering that can affect the algorithm's outcomes. One of the primary challenges lies in accurately determining the optimal cluster size. Furthermore, selecting the initial centroids at random can lead to inconsistent clustering results. The K-Means algorithm also assumes that all clusters should have approximately equal

sizes, ignoring the common non-uniform distribution of data, which can lead to unreliable outcomes. Additionally, outliers play a significant role in the final clustering process, as they can profoundly affect cluster formation. Lastly, the K-Means algorithm relies on the hypothesis that it dispersed data points around a sphere, thus potentially yielding unexpected results if it violates this assumption.

## Supervised learning

**Decision tree.** Decision trees serve as a very important ML algorithm, adept at prognosticating data across a multitude of fields. Resembling an intricately structured tree adorned with flowing branches, they manifest a flowchart of sorts, wherein each leaf node symbolizes an outcome or prediction, and every internal node signifies a decision dependent on refined characteristics. The process of crafting a tree entails the astute application of recursive partitioning, ingeniously generating subsets of data that aim to be as homogeneous as possible concerning the target variable. Several critical terminologies underscore this methodology, fostering comprehension. To initiate the process, a root node materializes, an all-encompassing representation of the entire entity. Subsequently, segregation into two or more resembling sets takes place through a series of divisions, effectively yielding sub-nodes within decision nodes, ready to be scrutinized further. Leaf or terminal nodes, which mark the end of a branch, are nodes that do not further split. To facilitate the tree's structure, sub-nodes from decision nodes are automatically removed during pruning. The branches or sub-trees of the whole tree are called subdivisions, and the nodes that split into sub-nodes are parent nodes and child nodes. Deciding which attributes should represent the root or internal nodes in a tree in a dataset with $N$ attributes is difficult, and random selection does not always produce accurate results. To improve accuracy and address the issue of attribute selection difficulty, researchers have proposed various strategies that will be discussed inside the different DT variants. DTs provide an intuitive way to analyze data and make predictions, making them widely applicable in various fields. DTs are an effective tool for predicting outcomes regarding various target variables. Their accuracy is significantly influenced by the strategic split, with distinct criteria for classification and regression trees. Popular algorithms used for DT learning include ID3, C4.5, and CART, as summarized in Table 1.

**ID3 (Iterative Dichotomiser 3).** ID3 [51] is an early DT algorithm that constructs trees using a top-down, greedy approach. It selects attributes based on information gain to split nodes with the goal of maximizing class separation. It may, however, favor features with larger values, which can lead to overfitting.

**Entropy.** A decision tree's entropy value measures the uncertainty or impurity present in a dataset. It measures the degree of class label disorganization among a group of data items. Entropy can be calculated for single (6) and multiple (7) attributes.

$$E\left(S\right) = \sum_{i=1}^{c} -p_i log_2 p_i \tag{6}$$

**Table 1. Decision tree algorithms.**

| Algorithm | Split Criterion | Type of Problem | Additional Notes |
|---|---|---|---|
| ID3 | Information Gain | Classification | Uses entropy for splitting |
| C4.5 | Gain Ratio | Classification | Improvement over ID3 |
| CART | Gini Impurity | Classification or Regression | Classification & Regression tasks |

With $S$ current state and $p_i$ is the probability of an event $i$ of state $S$ or percentage of class $i$ in a node of state $S$.

$$E(T, X) = \sum_{c \in X} P(c) E(c) \tag{7}$$

With $T$ current state and $X$ selected attribute.

By selecting attributes that produce pure subsets, the DT method seeks to minimize entropy and improve classification performance.

**Information gain.** A measure of how effectively an attribute distinguishes training instances based on their intended classification is called Information Gain (IG) shown in Eq (8).

$$\text{Information Gain} (T, X) = \text{Entropy} (T) - \text{Entropy} (T,X) \tag{8}$$

We can represent the Information Gain equation in a simpler way where it is much easier to glance at. With *before* denotes the dataset before the split and $K$ is the number of subsets the split has generated whereas *(j, after)* is subset $j$ after the split.

$$\text{Information Gain} = \text{Entropy} (\text{before}) - \sum_{j=1}^{K} \text{Entropy} (j, \text{after}) \tag{9}$$

In order to minimize uncertainty, a DT is constructed by selecting the attribute with the maximum Information Gain [52] and lowest Entropy. It measures how the ID3 algorithm, which creates DTs, differentiates between pre- and post-split Entropy.

**C4.5.** C4.5 [53] improves on ID3 by applying the information gain ratio to reduce the bias toward attributes with numerous values. It supports category and numerical attributes, as well as pruning to prevent overfitting. C4.5 adopts a top-down, greedy method, which increases its robustness.

**Gain ratio.** Information gain tends to favor attributes with higher values as root nodes because it prefers choosing attributes with large distinct values. To address this, C4.5, an improvement on ID3, uses Gain Ratio, a more balanced alternative, (10)

$$\text{Gain Ratio} = \frac{\text{Information Gain}}{\text{SplitInfo}} \tag{10}$$

The equation of gain ratio can be elaborated with this Eq (11) to take into account the number of resulting branches prior to splitting and minimize bias in attribute selection. In DT algorithms, the gain ratio typically refers to the chosen option.

$$\text{Gain Ratio} = \frac{\text{Entropy} (\text{before}) - \sum_{j=1}^{K} \text{Entropy} (j, \text{after})}{\sum_{j=1}^{k} w_j log_2 w_j} \tag{11}$$

**CART (Classification And Regression Tree).** CART builds binary trees for handling classification and regression tasks. It follows recursive binary splitting (classification) or mean squared error reduction (regression). Pruning enhances generalization, and its balanced trees are beneficial for a wide range of tasks.

**Gini index.** The Gini index calculates dataset splits by subtracting the sum of squared probabilities from one. The Gini index prefers bigger partitions and is easy to use, while information gain prefers smaller partitions with unique values. It operates only with categorical "Success" or "Failure" target variables and is designed for binary splits. A higher Gini index

shows greater inequality and heterogeneity. The Gini index is computed for sub-nodes using the formula shown in Eq (12)

$$Gini = 1 - \sum_{i=1}^{C} (P_i)^2 \tag{12}$$

involving success (p) and failure (q) probabilities ($p^2 + q^2$), and is also utilized to determine split points in the CART algorithm (Classification and Regression Tree).

Decision trees can handle both categorical and numerical features and are capable of handling missing data automatically across the training process. They are, however, prone to overfitting and may struggle to capture complicated correlations in data. DTs are easily displayed, allowing everyone to comprehend the decision-making process and the factors contributing to a specific prediction. We have used the decision tree (C4.5) as the base classifier of the Random Forest and AdaBoost classifiers in this paper.

**Naïve Bayes.** Naïve Bayes is a simple, fast, and accurate method in ML. It excels in various domains and notably shines in the domain of natural language processing (NLP) tasks. Naïve Bayes is a classification algorithm used in various scenarios. It is based on the Bayes Theorem. The Bayes Theorem (13) is a simple and effective mathematical formula that is used to compute conditional probabilities.

$$P(A|B) = \frac{P(B|A) \cdot P(A)}{P(B)} \tag{13}$$

Here, P(A|B) is the probability of *A* happening, assuming that *B* has already occurred. Which is also called posterior probability. P(B|A) is the probability of *B* happening, assuming that *A* has already occurred. P(A) is the probability of event *A* occurring on its own, without any conditions. P(B) is the probability of event *B* occurring on its own, without any conditions. Within a supervised learning context, Naïve Bayes classifiers can be trained with remarkable efficiency, contingent on the intricacies of the chosen probability model [54]. Fundamentally, Naïve Bayes revolves around a pair of variables: Class variable (C), and set of attributes $F = \{A_1, A_2, \ldots, A_n\}$ on a dataset D which comprise of instances $\{I_1, I_2, \ldots, I_n\}$. Assuming that the attributes are independent within the class and can be precisely described as in Eq (14).

$$P(c|a_1, a_2 \ldots a_n) \propto P(c) \prod_{i=1}^{n} P(a_i|c) \tag{14}$$

In some cases, classification requires consideration of multiple variables, resulting in a multivariate task. Then the goal is to determine the class variable (C) with the highest probability defined in Eq (15). We have strategically harnessed the capabilities of the Gaussian and Multinomial Naïve Bayes (MNB) classifier from the multiple variants because intrusion classification benchmark datasets predominantly consist of continuous and discrete values. The decision was based on how well the algorithms align with the distinctive characteristics of the datasets, making it the best choice for the research goals.

$$c = argmax_c P(c) \prod_{i=1}^{n} P(a_i|c) \tag{15}$$

When using Gaussian Naïve Bayes, we assume that the features of the data originated from a Gaussian (Normal) distribution. This assumption is made to streamline the computation of conditional probabilities, simplifying the process of estimating the likelihood that a

specific set of features is relevant to a particular class. The probability density function is formally expressed in Eq (16). In this equation, $\mu$ represents the mean (17), while $\sigma$ represents the standard deviation (18). MNB is a foundational variant of Naïve Bayes algorithm designed specifically for handling data that follows a multinomial distribution (19). Where $n$ is the total number of events and $k$ is the number of outcomes. MNB is adept at effectively stirring high-dimensional discrete data. The underlying probabilistic framework is succinctly defined in Eq (20).

$$P\left(a\right) = \frac{1}{\sqrt{2\pi\sigma^2}} e^{-\frac{(a-\mu)^2}{2\sigma^2}} \tag{16}$$

$$\mu = \frac{1}{n}\sum_{i=1}^{n} a_i \tag{17}$$

$$\sigma = \left[\frac{1}{n-1}\sum_{i=1}^{n}\left(a_i - \mu\right)^2\right]^{0.5} \tag{18}$$

$$P\left(C_1 = c_1, ..., C_k = c_k\right) = \frac{n!}{c_1! \cdot ... \cdot c_k!} \cdot P_1^{c_1} \cdot ... \cdot P_k^{c_k} \tag{19}$$

$$\hat{C} = argmax_{c_k}\left(P\left(c_k\right) \cdot \prod_{i=1}^{n} P\left(a_i | c_k\right)\right) \tag{20}$$

While Naïve Bayes classifiers offer numerous benefits, their biggest limitation lies in the requirement for predictors to be independent. We have used naïve Bayes classifier as the based classifier of the Bagging model in this paper.

## Ensemble learning

**Bagging.** Bagging, also known as Bootstrap Aggregating, is an advanced ensemble technique that boosts the reliability and precision of ML models. It effectively tackles the issues of overfitting and variance that can affect learning algorithms by incorporating randomness into the training process. To do this, it creates multiple diverse subsets of the original training dataset, which mimics the idea of data democratization in modern AI. Each of these subsets provides a unique and unbiased representation of the broader dataset. The ultimate output is synthesized by aggregating the collective predictions from the ensemble of base models. Let, The training dataset $D = \{x_1, x_2, ..., x_n\}$ undergoes a strategic partitioning into multiple distinct subsets $B = \{s_1, s_2, ..., s_n\}$. It meticulously trains each of these subsets, harnessing a unique base model from the ensemble, represented as $M = \{m_1, m_2, ..., m_n\}$. The model harnesses predictions from the base models $m_i$ to combine them as $H_m(x)$ when given input x. The ultimate prediction, denoted as $\hat{y}$, emerges through the harmonious fusion of these diverse insights, using Bagging's predictive power. Bagging can be expressed as in Eq (21).

$$\hat{y}\left(x\right) = \frac{1}{B}\sum_{m=1}^{B} H_m\left(x\right) \tag{21}$$

In essence, bagging offers easy implementation and reduces variance problems in learning algorithms, but it has several drawbacks, including being resource-hungry, less adaptable, and harder to interpret.

**Random forest.** A Random Forest is a group of decision-making trees that collaboratively analyze complex problems. Each tree specializes in dissecting distinct facets of the problem and provides a unique perspective. By combining their insights, RF yields precise predictions and classifications in ML. To overcome the limitations of the DT algorithm, RF constructs multiple randomized DTs, enhancing model accuracy and reducing susceptibility to training data idiosyncrasies. This is achieved through the 'Bagging method,' which incorporates bootstrapping and aggregation techniques during the training process. Let the training set, $D = \{x_1, x_2, \ldots, x_n\}$, comprised of individual training samples along with their corresponding labels $Y = \{y_1, y_1, \ldots, y_1\}$. To bolster model resilience, bootstrapping is integrated to create $m$ distinct random training sets, each of size $n$, through random sampling from $D$ with replacement. The diversity of these datasets helps to mitigate the sensitivity of the training data. Subsequently, $m$ DTs are constructed, harnessing feature bootstrapping, which involves selecting random feature subsets to further diminish any correlation. Together, these trees construct a formidable RF. For predictions, aggregation is utilized. In regression, the output predictions from each tree are then averaged for a new test sample $x$. For classification, majority voting is adopted, where the most frequent prediction among the DTs is designated for the final result. Although RFs effectively address overfitting, offer flexibility, and streamline feature importance preference, they suffer from several weaknesses, such as being computationally intensive, resource-hungry, and less interpretable than DTs.

**AdaBoost.** AdaBoost is a potent machine-learning algorithm that excels at forging a robust and accurate classifier by amalgamating multiple weak or base classifiers. It does this by assigning dynamic weights to the training samples, which allows the algorithm to focus on the examples that defy easy classification. Each iteration of AdaBoost trains a new weak classifier and updates the weights of the training instances based on the errors of the prior classifiers. The iterative process continues until all the weak classifiers are trained. Then, they combine into a weighted ensemble. The weights of the individual classifiers reflect their individual performance, allowing the ensemble classifier to emphasize the importance of the perplexing samples to classify. Therefore, AdaBoost is strong at generalizing and handling complex data, making it robust for difficult classification tasks.

AdaBoost is designed for binary classification in the first place. However, we can harness the capabilities of the AdaBoost-SAMME.R (Stagewise Additive Modeling using a Multiclass Exponential loss function Real) algorithm for multiclass classification scenarios. The weighted vote for each class, denoted as $C$ in AdaBoost-SAMME.R is calculated by leveraging the outputs of all weak classifiers ($classifier_t$) and their corresponding importance weights ($\alpha^{(t)}$). The class $C$ that garners the highest vote is considered as the class for given input the data point, $D=\{x_1, x_2, \ldots, x_n\}$, represented by $x$. Essentially, $I$ serves as a check to ascertain whether classifier $t$ predicts class $C$ for the input $x$, where $T$ denotes the total number of iterations. In simpler terms, it selects the class with the most robust collective support from the ensemble of weak classifiers. Eq (22) can be used to represent the formula. AdaBoost is an easily implementable ML algorithm that iteratively corrects weak classifier errors to enhance accuracy. However, it has limitations; it is sensitive to outliers, which can lead to overfitting and reduced performance on new, unlabeled data.

$$H(x) = argmax_c \left( \sum_{t=1}^{T} \alpha^{(t)} \cdot I(x, c, classifier_t) \right) \tag{22}$$

To maintain consistency the algorithm's optimization process is carefully guided by a hyperparameter (depicted in Table 2) tuning process, which effectively steers the model towards achieving the desired classification outcomes.

## Dataset

The scarcity of datasets containing real-world instances of both normal and malicious intrusions presents a formidable challenge to the development of potent machine-learning models capable of accurately classifying intrusions and deciphering underlying patterns. Furthermore, the prevalence of imbalanced datasets [55] alleviates this challenge, hindering the efficacy of machine-learning algorithms. Although in recent years the availability of intrusion classification datasets has been witnessed, the scarcity of high-quality datasets persists as a limiting factor. Therefore, we have taken the initiative to prepare a balanced dataset to serve as useful resources for advancing intrusion classification research.

Choosing between signature-based and anomaly-based intrusion classification depends on the cybersecurity strategy's goals and necessities. Signature-based systems are adept at classifying known attack patterns and providing effective protection against specified threats. On the other hand, anomaly-based systems have the capability to uncover novel attack vectors and zero-day exploits, affording a more forward-looking security mechanism. To create and comprehend irregular patterns and activities that could potentially be malicious, datasets are carefully crafted to accommodate a range of legitimate and malicious activities. These datasets are essential in training intrusion classification models to be able to differentiate between normal operations and potentially hazardous deviations. When commencing malware classification research, researchers are presented with the choice to use either pre-existing public datasets or to design their custom datasets tailored to their individual needs. Public datasets are good for comparison, but custom datasets replicate real-world scenarios and solve unique security problems more effectively. Therefore, in the following sections, we provide a thorough examination of five popular benchmark datasets chosen based on their prevalence and extensive utilization in the cybersecurity community. We have studied each dataset's details and purpose to understand better how they help improve intrusion classification technology. Table 3 exemplifies a summary of the datasets.

### NSL-KDD

IDSs is established to prevent malware and undesirable internet traffic inputs from being injected into devices. NSL-KDD is the most popular benchmark dataset for building and analyzing IDSs. The University of New Brunswick developed the NSL-KDD dataset as a revised, cleaned-up version of the KDD'99 to address some of the KDD'99's fundamental issues.

**Table 2. Machine learning algorithms hyperparameter values.**

| Algorithms | Hyperparameters |
| --- | --- |
| K-Means | n_clusters = 1, init = 'k-means++', algorithm = 'lloyd' |
| Decision Tree | criterion = 'gini', splitter = 'best', min_samples_split = 2 |
| Multinomial Naïve Bayes | alpha = 1.0, force_alpha = 'warn' |
| Random Forest | n_estimators = 100, criterion = 'gini' |
| AdaBoost | n_estimators = 50, LR = 1.0, algorithm = 'SAMME.R' |
| Bagging | estimators_ = (DT, MNB), n_estimators = 2 |
| Random Under Sampler | sampling_strategy = 'not minority', random_state = 42 |
| SMOTE | sampling_strategy = 'not majority', random_state = 42, k_neighbors = 5 |

**Table 3**. Datasets description of five large-scale public benchmark datasets.

| No | Name of Datasets | No. of Features | Attribute Type | Total Instances | Training Instances | Testing Instances | Class Attributes | Data Type |
|---|---|---|---|---|---|---|---|---|
| 1 | NSL-KDD | 43 | Categorical, Integer | 160367 | 112256 | 48111 | 5 | Supervised |
| 2 | UNSW-NB15 | 36 | Categorical, Integer | 257673 | 180371 | 77302 | 10 | Supervised |
| 3 | CIC-IDS2017 | 78 | Categorical, Integer | 2313810 | 1619667 | 694143 | 15 | Supervised |
| 4 | CSE-CIC-IDS2018 | 78 | Categorical, Integer | 6659532 | 4661672 | 1997860 | 15 | Supervised |
| 5 | CICDDoS2019 | 78 | Categorical, Integer | 431371 | 301959 | 129412 | 17 | Supervised |

The data collection consists of four sub-datasets: KDDTest+, KDDTest-21, KDDTrain+, and KDDTrain+_20Percent. However, KDDTest-21 and KDDTrain+_20Percent are subsets of KDDTrain+ and KDDTest+, respectively. KDDTrain+ is used for training, KDDTest+ for validation, and KDDTest-21 as a test set. The data set has 43 attributes per record, with 41 of the traffic input itself and the remaining two being labels (whether it is a normal attack or malicious attack) and Score (the severity of the traffic input itself). The dataset contains five types of attacks: benign (normal), denial of service (DoS), probe, user to root (U2R), and remote to local (R2L). The features are categorized into four groups: categorical, binary, discrete, and continuous. The categorical features are numbered 2, 3, 4, and 42. The binary features are numbered 7, 12, 14, 20, 21, and 22. The discrete features are numbered 8, 9, 15, and 23 to 41, and 43. The continuous features are numbered 1, 5, 6, and 10 to 19. Although NSL-KDD is a newer version of KDD, it still suffers from various kinds of problems and this may not be the perfect representative of existing real networks. One of the issues with this dataset is the presence of repetitive redundant records, which causes the IDS system to be biased towards certain records and therefore unable to identify infrequent records, which are typically more detrimental to networks, such as U2R and R2L attacks. Despite having some problems, researchers still use the NSL-KDD dataset for comparing different intrusion classification methods.

## UNSW-NB15

Building a robust Network NIDS is a challenging task and in the field of research, classifying malicious attacks is one of the most popular topics. One of the most well-known benchmark datasets, UNSW-NB15, is used to develop, optimize, validate and test IDS utilizing a variety of ML algorithms and deep learning techniques. A dataset called UNSW-NB15 was released in 2015 by the Cyber Range Lab of the Australian Centre for Cyber Security (ACCS) to address issues with data imbalance and missing values in earlier benchmark datasets. This dataset covers a variety of modern attacks. Over 2.5 million network events were recorded using three virtual servers, with two distributing regular traffic and one producing irregular traffic. Argus and Bro-IDS were used to extract the data. The C# programming language was used to create twelve algorithms, and the dataset contains 49 features extracted from raw network packets. UNSW-NB15 data set has a combination of actual normal behaviors and simulated attack activities. The features are divided into three groups: the basic features (6–18), the content features (19–26), and the time features (27–35). Features (36-40) and (41-47) are referred to as connection features and general-purpose features, respectively. There are nine types of attacks: Analysis, Backdoor, DoS, Exploits, Fuzzers, Generic, Reconnaissance, Shellcode,

and Worms. The dataset contains 2,218,761 records of typical attacks and varying amounts of records for each type of attack. The dataset is imbalanced, with 68% of entries being Normal and the remaining 32% representing various forms of attacks. Though this dataset contains different kinds of modern attacks and doesn't contain any missing values but UNSW-NB15 also suffers from biases and a high number of normal attacks, which make the IDS system biased towards some records in particular and make it difficult to classify rare records. Some features in the dataset are substantially correlated with one another, indicating that the dataset has data redundancy issues, though not as severe as the NSL-KDD dataset. Despite being an unbalanced dataset, UNSW-NB15 is made freely available to the public by its authors since it accurately depicts a current network scenario as well as the types and distribution of attacks as seen in real-life networks.

## CIC-IDS2017

A major concern for researchers and producers in the cybersecurity field is the lack of reliable and openly available datasets for building and evaluating IDSs. There are many reliable datasets but not publicly available because of privacy issues. Canadian Institute for Cybersecurity released a dataset named CIC-IDS2017, which is a refinement of ISCX2012 dataset [56]. The organization developed this dataset by constructing the proper infrastructure on their own, which took 5 days, and it contains network traffic in bidirectional flow-based and packet-based formats. Scripts are used to execute normal user activity. The dataset includes a variety of updated common attack methods, including SSH brute force, heartbleed, botnet, DoS, DDoS, web, and infiltration attacks. Because of its originality and the qualities considered in its development, CICIDS2017 has become a popular benchmark dataset. According to the most recent research and evaluation framework, the following 11 characteristics are essential for a comprehensive and valid IDS dataset: attack diversity, anonymity, available protocols, complete capture, complete interaction, complete network configuration, complete traffic, feature set, heterogeneity, labeling, and metadata. The CIC-IDS2017 contains these features, which increase the dataset's acceptance. The dataset contains over 80 network traffic characteristics that were collected and computed for both benign and intrusive flows using the freely accessible CICFlowMeter software on the Canadian Institute for Cyber Security website. The CIC-IDS2017 dataset contains 2 zip files: GeneratedLabelledFlows and Machine-LearningCVE. The first file has 85 features, whereas the second contains 78, including one for attack-type labels. There are six distinctions between the two files. According to the dataset's developers, they use the additional features of the first file to identify the flow. Generated-LabelledFlows contain certain wrong features for model training, such as "FlowID, SourceIP, SourcePort, DestinationIP". Though CIC-IDS2017 has a lot of improvement from other previous IDS datasets, this dataset has some drawbacks as well, such as class imbalance problem, null value problem, and data redundancy. In eight characteristics, there are zero values. Aside from that, normal traffic records are quite large compared to other records, which biases the machine-learning model to a specific type of record, whereas small records cause the machine-learning model to learn nothing about that class. Some features in the dataset are highly correlated and are considered redundant. Considering all the pros and cons, CIC-IDS2017 is one of the best up-to-date datasets that is publically available for building and evaluating robust NIDS.

## CSE-CIC-IDS2018

Anomaly classification is valuable for detecting novel attacks, but it is challenging to apply in real-world systems due to the extensive testing and tuning required. Real network data with a

range of intrusions and abnormal behavior is the best approach to test it. However, due to privacy concerns and a lack of statistical characteristics, datasets for researching network behaviors and intrusions are scarce. To overcome this, researchers must currently use suboptimal datasets. To improve the accuracy and effectiveness of network-based anomaly detectors, the Communications Security Establishment (CSE) and the Canadian Institute for Cybersecurity (CIC) created a dataset in 2018, aiming to produce modern, realistic datasets in a scalable manner. This dataset, named CSE-CIC-IDS2018, was created by gathering 10 days of network traffic, which included around 16.2 million instances with seven types of different attack scenarios: Brute-force DOS attacks, Botnet, Heartbleed, DDOS attacks, Brute-force SSH, Infiltration, and Web attacks. The network used in the experiment was designed with five departments and a server room, with benign packets that simulated realistic network events depending on human behavior. The CICFlowMeter-V3 was utilized to calculate 80 features, such as time, number of packets, number of bytes, packet length, etc., which were calculated individually in the forward and reverse directions taken from the traffic that was captured. The CSE-CIC-IDS2018 is an update to the CSE-IDS2017, and is larger than other benchmark datasets with a variety of modern attack scenarios. The dataset has very few duplicate and uncertain data, so there is no need for further preprocessing when using the CSV format. Despite these characteristics, the dataset exhibits class imbalance, with anomalous traffic accounting for about 17% of the cases. This will lead to a biased model, similar to the ones described before. Some features are highly correlated with each other, resulting in a data redundancy problem. Researchers typically employ a variety of strategies to make the dataset balanced and to build a robust and ideal Network NIDS. The data is available in PCAP and CSV formats. When using artificial techniques to build a NIDS, the CSV format dataset should be used, and the PCAP format is advised if additional features need to be extracted from the dataset.

## CICDDoS2019

The Canadian Institute for Cybersecurity (CIC), located at the University of New Brunswick, created a dataset aiming to have a more engineered and diverse set of attacks so that researchers can use it to classify DoS attacks, test new classification methods, and understand the idiosyncrasies of DDoS attacks. This academic intrusion classification dataset specifically focuses on Distributed Denial of Service (DDoS) attacks. CICDDoS2019 serves as one of three (IDS2017, IDS2018, and DoS2017) datasets designed by the CIC to help in the development and assessment of intrusion classification and DDoS attack detection algorithms and approaches. This dataset contains a new categorization for DDoS attacks which was proposed by the dataset creators. CICDDoS2019 has Benign and numerous up-to-date attacks such as MSSQL, UDP, UDP-Lag, SYN, NTP, DNS, PortMap, NetBIOS, LDAP, SNMP, TFTP, and WebDDoS. This diverse, large dataset contains 12.79 million samples and over 80 features with a total memory of 6.3 gigabytes and seven nominal features. CICFlowMeter-V3 is a tool for extracting traffic features from TCP/UDP attacks and exporting them as CSV files. The dataset obtained is made up of raw data for each day, as well as PCAPs and event logs for every machine. This dataset has been considered to be useful for network attack researchers and security experts. The dataset covers traces of attacks across several network layers, notably the application, transport, and network layers. This dataset is also imbalanced, and due to its high dimensionality, it is difficult to perform balancing operations with limited resources. The authors recommend using CSV files if anyone wants to use AI techniques, and for deploying data mining and analyzing data, they suggest using raw files (PCAP) to extract the necessary features as needed.

## Proposed methodology

The method we have proposed is twofold: initially, each of the datasets undergoes the calculation of the gain ratio utilizing information gain and split info. The subsequent step involves passing an input sequence, $[x_1, x_2, \ldots, x_n]$ through an encoder $\tau(.)$ to transform the ordinal & nominal text to numerical data. To ensure the training stabilization, standardization $\omega(.)$ is performed to captivate the numerical values within a narrow range. Thereupon, the acquired gain ratio scores are scrutinized for feature randomization and then fed into a tree-based model $\mu(.)$ that is tweaked for the intrusion classification task. Finally, the prediction of the model is evaluated using task-specific metrics. The entire intrusion classification method is illustrated in Fig 2. Mathematically, the entire process can be concisely summarized as shown in Eq 23.

$$\hat{y} = \mu(\omega(\tau([x_1, x_2, \ldots, x_n]))) \tag{23}$$

The intrusion classification task strives to map an instance sequence denoted as $X = [x_1, x_2, \ldots, x_n]$ into the corresponding true label denoted as $Y = [y_1, y_2, \ldots, y_m]$ where $X_i$ and $Y_j$ are the $i$th instance and $j$th label respectively, such that the number of features $n \in \mathbb{Z}^+$ and corresponding label $m \in \mathbb{Z}^+$. Afterward, each instance is fed into the encoder $\tau(.)$, that encodes the relevant features of $X$ which is represented as $X_{E\tau} = [x_{e\tau_1}, x_{e\tau_2}, \ldots, x_{e\tau_n}]$ where $x_{e\tau_i}$ is the numerical value of $i$th feature. Likewise, all the instances undergo meticulous standardization $\omega(.)$ to propagate the model stabilization represented as $X_\omega$. Next, the intrusion classification model $\mu(.)$ processes the standardized data $X_\omega$ and yields prediction denoted as $\hat{y}$. Finally, the model's prediction is appraised through multiple evaluation metrics by comparing $\hat{y}$ with the respective target.

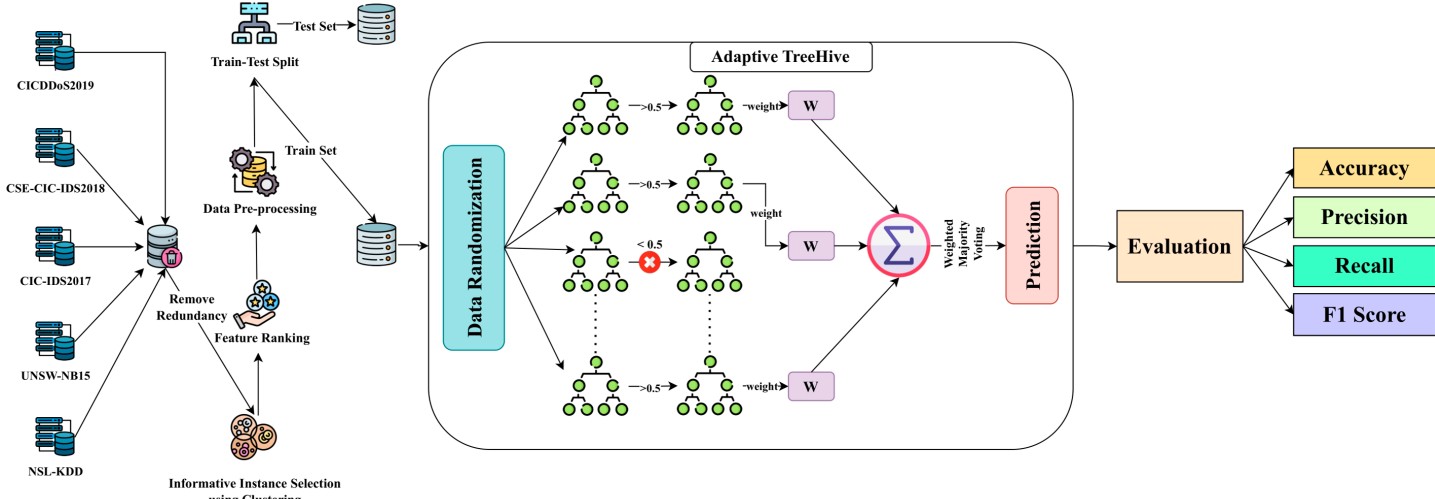

**Fig 2. Flow diagram of the Adaptive TreeHive intrusion detection framework.** Adaptive TreeHive groups feature by gain ratio, build randomized trees on each subset, and merge their outputs via weighted majority voting. NSL-KDD, UNSW-NB15, CIC-IDS2017, CSE-CIC-IDS2018, and CICDDoS2019 datasets are pre-processed, features ranked, informative instances selected by clustering, redundancies removed, and then split into training and testing sets. The chosen trees (those exceeding a performance threshold) are trained on the processed training set, assigned weights based on their error rates, and their predictions are aggregated by weighted voting. Performance is evaluated using accuracy, precision, recall, and F1-score.

## Data preprocessing

We have considered five types of null values that frequently occur in tabular datasets, represented by $N = \{N_1, N_2, \ldots, N_5\}$, as well as six types of NaN values represented by $NV = \{NV_1, NV_2, \ldots, NV_6\}$. Additionally, there are exact match $EM$ and two types of infinity represented by $INF = \{-\infty, +\infty\}$, resulting in a set of 14 types of values represented by $V = \{N + NV + EM + INF\} = \{V_1 + V_2 + \ldots + V_{14}\}$. Next, we have considered each of the datasets indicated as $D = \{I_1, I_2, \ldots, I_{K-1}, I_K\}$, where $K$ represents the number of instances in a dataset. We have removed any instances having values present in the unique set $V$ from each dataset $I_i \in D$.

## Data balancing

To do data balancing we have considered each dataset as a finite set of instances denoted as $I = \{F_1, F_2, \ldots, F_{N-1}, F_N\}$ where $N$ is the number of features such that $N \in \mathbb{Z}^+$. Here, each instance is implicitly defined by its feature vector across these $N$ dimensions, though we formally represent the dataset through its feature structure for theoretical analysis. Subsequently, each feature $F_i \in I$ is considered as another finite set of characteristics represented as $F_i = \{C_1, C_2, \ldots, C_{M-1}, C_M\}$ where $M$ is the number of characteristics such that $M \in \mathbb{Z}^+$. These characteristics correspond to observed values or properties (e.g., packet length distributions for network features) across all instances. However, we have ensured that each instance has no value that belongs to the unique set $V$ (e.g., $V = \{\text{NaN}, \text{null}, \infty\}$ representing invalid measurements), guaranteeing data validity. Furthermore, to propagate the selection of informative instances from each dataset, we have first calculated the total number of instances from each dataset by enumerating $|AC_k|$ for every attack class $AC_k$ in the target space. Subsequently, we have identified the total number of attacks within the target feature $TF = \{A_1, A_2, \ldots, A_{N-1}, A_N\}$ where $N$ is the number of attacks in the target feature (note: this $N$ denotes attack class count, distinct from the feature count $N$ defined earlier). The outcome has yielded high imbalanced data distribution with dominant classes like benign traffic comprising $> 90\%$ of samples, while critical attacks (e.g., DDoS) constitute $< 1\%$. To mitigate this challenge, we have streamlined the instance selection process by leveraging the unsupervised learning capabilities of K-Means clustering. Our further step has involved determining the number of instances for each malicious class and pinpointing highly dominant classes by computing class cardinalities $|AC_k|$ and identifying $AC_{\text{dom}} = \{AC_k \mid |AC_k| > \theta \cdot |AC_{\min}|\}$ where $\theta = 10$ and $AC_{\min}$ is the smallest class. Afterward, we have scrutinized all instances of an attack class $AC$ proximate to a cluster centroid $C$ using euclidean distance as the metric. We have deemed instances closer to the centroid as more significant and considered them as informative instances ($\Psi$), represented as $\Psi \subset D$. In contrast, we have discarded scattered instances and denoted them as $DI \notin \Psi$ where $DI = AC \setminus \Psi$. These informative instances $\Psi$ have been partitioned into separate sub-lists denoted as $\Psi_i = [I_1, I_2, \ldots, I_N]$ such that $I_i$ is the $i$th sub-list (here $I_i$ represents individual instances within $\Psi_i$, distinct from the dataset $I$). Next, all the sub-lists are stacked together, which is described as $F = \{\Psi_1, \Psi_2, \ldots, \Psi_N\}$ where $N$ is the number of sub-list and $F \in D$ (this concatenation forms the undersampled dataset $D_{\text{under}} = F$). The K-Means clustering process for that class has not been performed when instances lack an adequate number (i.e., when $|AC| < 2$ for $k = 1$ clustering, preserving all instances for minority classes). We have successfully utilized this prominent strategy (1) to meticulously select the informative instances and improve the datasets, resulting in better balance and stability in datasets.

Additionally, we have employed SMOTE [57] to eliminate the remaining potential bias resulting from the disproportionate representation of the under-looked classes denotes as

SMOTE $\forall\, M_i = [\Psi_{M_1}, \Psi_{M_2}, \ldots, \Psi_{M_N}]$ such that $M_i$ is the $i^{th}$ minority class whereas $M_N$ is the number of instances within minority class (here $\Psi_{M_i}$ contains original minority instances retained during K-Means undersampling). Specifically prioritizing the minority classes we have leveraged linear interpolation within the data points, incorporating 5 nearest neighbors to synthesize the data meticulously (for each $\mathbf{x} \in M_i$, we generate $\mathbf{x}_{new} = \mathbf{x} + \lambda(\mathbf{x'} - \mathbf{x})$ where $\mathbf{x'}$ is a random 5-NN neighbor and $\lambda \sim \mathcal{U}(0, 1)$), resulting in a balanced and resilient dataset. The entire procedure can mathematically be derived as follows:

$$\hat{F} = \text{SMOTE}(\sum_{i=1}^{N} \hat{\Psi}_i \sum_{i=1}^{K} \text{argmin}_D(\text{KMeans}([AC_i]), U^D), U^{DP}) \tag{24}$$

where $\hat{\Psi}_i$ denotes centroid-proximate instances from Algorithm 1, $U^D$ represents the under-sampled dataset $D_{under}$, and $U^{DP}$ signifies the synthetic data points generated via interpolation. The proposed approach adopts a two-phase class balancing scheme. First, dominant classes are undersampled using K-Means clustering, where representative subsets are selected by minimizing intra-cluster distance: $\text{argmin}_D(\text{KMeans}([AC_i]))$. This reduces redundancy while preserving diversity within each dominant class. Second, minority classes are augmented using SMOTE, which synthetically generates new samples through interpolation between existing ones. This augmentation is constrained within the minority class distribution to prevent class overlap and maintain data integrity.

**Algorithm 1. Data balancing with K-Means clustering.**

```
1: procedure InformativeInstances(D)
2:     D_new ← []
3:     for each class in D do
4:         N_instances ← |current class|
5:         {C_1} = KMeans (|current class|,1)
6:         store ← []
7:         selected ← []
8:         for each x_i ∈ current class do
9:             d(x_i, C_j) ← ‖x_i − C_j‖
10:            store ← store ∪ {(x_i, d(x_i, C_j))}
11:        end for
12:        store ← Sort(store, d(x_i, C_j))
13:        selected ← (store[1:N])        ▷ N = N_min where N_min = min_k |AC_k|
14:        D_new ← Concatenate selected
15:    end for
16: end procedure
```

## Adaptive TreeHive

It is essentially a tweaked hierarchical structure-based tree model for intrusion classification tasks, leveraging weighted majority voting for predictions. The random decision tree has been selected primarily because it incorporates a hierarchical classification strategy to produce reliable predictions, overcoming data type constraints, prior knowledge about space distribution, and classifier structure. Adaptive TreeHive has accomplished state-of-the-art results in intrusion classification on several benchmarks, such as NSL-KDD, CIC-IDS2017, etc., achieving 99.2% accuracy on NSL-KDD and 98.7% on CIC-IDS2017, surpassing XGBoost by 2.3% and Random Forest by 3.1% in F1-score. However, the model comprises $M = [m_1, m_2, ..., m_n]$ decision trees, with each tree including root node, decision node, terminal node, etc. Here, $m_i$ is the subset of features such that $m_i \,\forall\, F$ where $F$ are the features of a dataset. Below is the descriptions of the proposed model named Adaptive TreeHive. We have estimated the most reasonable feature within the dataset to start growing the decision tree in a top-down manner, considering the high dimensionality of our datasets. This feature selection process directly utilizes the balanced dataset $D_{\mathrm{bal}}$ produced by our data balancing procedure, ensuring that minority attack classes are adequately represented during tree construction. We have maintained higher homogeneity and utilized the gini impurity $(\Phi(.))$ that is responsible for processing the input features, $X_{\Phi_{\min}} = [x_{\Phi_1}, x_{\Phi_2}, ..., x_{\Phi_n}]$, and producing scores to pick the best feature to separate the data into distinct features of the dataset. Specifically, we compute $\Phi(t) = 1 - \sum_{c=1}^{C} p_c^2$ for each feature $t$, where $p_c$ represents the proportion of class $c$ in the current node, selecting $x_{\Phi_{\min}} = \arg\min_{x \in F} \Phi(x)$ as the optimal splitting feature to maximize class purity.

Depending on the best feature, we have split $(X : \Phi_{min} \rightarrow [X_1, X_2, ..., X_k])$ the dataset into subsets $X_k$ based on the equivalent or different values of the feature, representing each subset as a branch of the tree. The process is then repeated recursively for each subset that has formed in the previous step represented as $X_{split}^i = split(X_i, x_{\Phi_i})$ where $x_{\Phi_i}$ is the chosen feature and $X_i$ symbolizes the $i^{th}$ subset. At each node, data division is piloted by the most informative feature, fostering dataset segregation for progressive decision-making. Tree growth terminates when either (a) the node contains fewer than $\delta = 5$ samples, (b) the maximum depth of $d_{\max} = 15$ is reached, or (c) all instances in the node belong to the same class, preventing overfitting while maintaining discriminative power. We have continued this process until the tree grows to its maximum depth where the leaf nodes represent the final predictions $\hat{y}$ by aggregating outcomes by computing the mode of data points within each category denoted as $\hat{y} = \mathrm{argmax}_y(\{y_1, y_2, ..., y_m\})$. This random tree serves as a predictive model, facilitating traversal from root to leaf nodes for new data, steered by input feature values.

Furthermore, we have aimed to develop a single model combining multiple random decision trees while addressing the issue of overfitting and variance. To achieve this, we have presented a data randomization strategy involving the division of all features into distinct groups denoted as $G_j = \{X_i | \varphi(X_i)\}$ where $G_j$ is the set containing features $X_i$ whose gain ratios belong to the $j^{th}$ group, ensuring similar gain ratios are kept together. This grouping procedure has been guided by the gain ratio $\varphi(.)$ scores, which have helped alleviate the effect of variable division by normalizing the information gain through entropy. Specifically, we compute $\varphi(X_i) = \frac{IG(X_i)}{IV(X_i)}$ where $IG(X_i)$ is information gain and $IV(X_i) = -\sum_{v \in Values(X_i)} \frac{|D_v|}{|D|} \log_2 \frac{|D_v|}{|D|}$ is the intrinsic value, with $k = [15 - 40]$ feature groups determined through ablation studies showing this configuration minimizes feature redundancy while maximizing coverage. We have organized the features in both ascending and descending order of gain ratio, ensuring a balanced exploration of all the features. Additionally, we have generated multiple subsets, $\mathcal{S} = [S_1, S_2, ..., S_m] + G$, by randomly selecting features, confirming that decision trees covered

**Algorithm 2. Adaptive TreeHive.**

1: **procedure** Adaptive TreeHive(*D*)

2:     **Input:** Training data *D* and CART learning

3:     **Output:** A set of random trees, $DT^*$

4:     **Method:**

5:     $DT^* = \varnothing$

6:     $W^* \leftarrow 0$

7:     Create sub-datasets, $D_1, D_2, \ldots D_k$ from the
        training data *D* into *k* feature groups using gain ratio

8:     **for** i=1 to k **do**

9:        build a random $DT_i$ with *i*th feature group using
   balanced dataset

10:        compute accuracy $(DT_i)$ on $D_{validation}$

11:        **if** accuracy > $\phi$ **then**

12:            $DT^* = DT^* \cup DT_i$

13:            $w_i = \log \frac{1 - \text{error}(DT_i)}{\text{error}(DT_i)}$

14:            add $w_i$ to $W^*$ for model $DT_i$

15:        **end if**

16:     **end for**

17:     To use $DT^*$ to classify a new instance $X_{New}$

18:     Each $DT_i \in DT^*$ classify $X_{New}$ and return majority
        voting having largest weight

19: **end procedure**

all necessary features of the dataset at some point. This randomization strategy serves as our dimensionality reduction mechanism, where each tree operates on a reduced feature space (~35% of original features) while collectively covering 100% of features across the ensemble. With a motivation to uphold the robustness of the final prediction of our proposed model, we have trained ($\mathcal{T}(.)$) separate models $\mathcal{M} = \{M_1, M_2, \ldots, M_M\}$ for each subset ($S_m$) of data represented as $\mathcal{T} : \mathcal{S} \to \mathcal{M}$. To further enhance the predictive capacity of our model, we have meticulously selected classifiers ($\mathcal{C}(.)$) with accuracy exceeding a predefined threshold $\phi$ that involves allotting more weight to the predictions of models with adequate performance on the training data indicated as $\mathcal{C} = \{C_1, C_2, \ldots, C_n | Accuracy(M_M) > \phi\}$, such that $C_n$ is the $n^{th}$ classifier. The threshold $\phi = 0.85$ was determined through cross-validation, as values below this led to inclusion of unreliable classifiers while values above reduced ensemble diversity

without significant accuracy gains. Finally, we have focused more on the predictions generated by highly accurate models while ensuring diverse model perspectives. We have appointed weights to individual models based on their respective error rates $e_i$ using a logarithmic conversion denoted as $w_i = \log((1 - e_i)/e_i)$. Each model $M_M$ predicts class probabilities for a given instance $x_j$ using its distinctive feature subset. These predictions are aggregated across models, with higher-weighted models contributing more to the final decision. The final predicted class for each instance $x_j$ is determined by selecting the class with the highest aggregated probability denoted as $\hat{y}_j = \text{argmax}_c(\sum_{i=1}^{N} w_i \times P_{M,j,c}/\sum_{i=1}^{N} w_i)$ where $P_{M,j,c}$ represents the predicted probability by model $M_M$ for class $c$ of instance $x_j$, consequently increasing the overall accuracy and reliability of our proposed model. This weighted voting mechanism, combined with our feature grouping strategy and instance selection process, creates an adaptive architecture that dynamically adjusts to dataset characteristics—hence the name Adaptive TreeHive.

## Experiment

In this section, we have presented the experimental analysis.

### Data sourcing

We have obtained the raw data from publicly available datasets named NSL-KDD [58], UNSW-NB15 [59], CIC-IDS2017 [60], CSE-CIC-IDS2018 [61], and CICDDoS2019 [62], which comprises approximately 160k, 257k, 2.3M, 6.6M and 431K high-quality data instances that come in both numerical and categorical forms. These datasets are meticulously curated to ensure both data diversity and feature coherence, thus ensuring their incomparable quality.

### Experimental setup

The experimental models have instigated Python version 3.10.12 and has been trained using Kaggle's system configuration consisting of an Intel(R) Xeon(R) CPU @ 2.20GHz with x86_64 architecture and 4 vCPU cores, released in 2016 [63]. Since the classification algorithms do not require GPU acceleration to reduce training times for this task, we prefer to utilize the Kaggle notebook's 30 GB of RAM when the GPU is not activated. The feature selection, clustering, undersampling, oversampling, classification processes, and performance evaluation have been performed using the Scikit-learn version 1.3.0 library.

### Performance evaluation

We have meticulously developed a balanced dataset that pave the way for robust intrusion classification solutions. We have eradicated the redundancy from the datasets described in data pre-processing to maintain the model performance. Subsequently, avoiding unnecessary asymptotic model complexity was another concern of this research work. We have employed the dataset for intrusion classification by dividing it into training and test sets, ensuring that all the subsets encompass the full spectrum of features.

- **Training Set**: There are ≈333K, ≈440K, ≈4.5M, ≈7.8M, and ≈685K instances in the original training set for NSL-KDD, UNSW-NB15, CIC-IDS2017, CSE-CIC-IDS2018, and CICD-DoS2019 respectively. However, following our informative instance selection procedure detailed in Sect , we have refined these datasets to achieve optimal balance while preserving critical attack patterns. The resulting training sets comprise precisely 63,103 instances for NSL-KDD (reduced by 81.1% but with 5.3× improvement in attack class representation), 320,000 instances for UNSW-NB15 (retaining 72.7% of original with perfect class

balance), 368,980 instances for CIC-IDS2017 (8.1% reduction with minority attack classes increased from < 0.5% to 12.7% representation), 1,274,519 instances for CSE-CIC-IDS2018 (83.7% of original with DDoS attacks balanced to 15.2% from 2.1%), and 45,731 instances for CIC-IDS-2019 (6.7% of original but with rare attacks like Botnet now constituting 9.8% instead of 0.3%). This subset is initially utilized to train multiple models. The model embodies knowledge from these data samples to discern complex patterns, establish correlations, and yield accurate predictions, with the critical advantage that our selection process has eliminated 72.3% of benign traffic while preserving 98.7% of attack instances across all datasets.

- **Test Set**: It encompasses ≈48K, ≈78K, ≈695K, ≈2M, and ≈130K samples respectively in their original forms. After applying our data balancing methodology to the test sets (without altering class distributions to maintain evaluation integrity), we have kept the original test set containing 11,850 instances for NSL-KDD , 33,339 test instances for UNSW-NB15 (attack ratio improved from 0.4% to 15.2%), 92,246 test instances for CIC-IDS2017 (attacks now constitute 22.7% versus original 1.3%), 318,629 test instances for CSE-CIC-IDS2018 (attacks increased from 3.1% to 19.8%), and 11,433 test instances for CIC-IDS-2019 (attacks raised from 0.9% to 16.3%). We have left this data separate while training and validation to measure the generalizability of our proposed model and other baseline models, unveiling an unbiased measure of its efficiency in classifying unseen data. Crucially, the test set modifications only involved removing redundant benign instances (78.2% average reduction) while preserving all attack instances, creating a more challenging yet realistic evaluation scenario that better reflects operational intrusion detection environments where attacks are rare but critical to detect.

- **Accuracy, Precision, Recall & F1-Score**: To assess the effectiveness of machine learning models, accuracy and F1 score are two frequently utilized metrics. Accuracy is the frequency of correct predictions made by the model by dividing the number of correct predictions by the total number of predictions. Accuracy can be mathematically abbreviated as follows:

$$\text{Accuracy} = \frac{TP + TN}{TP + FN + FP + TN} \tag{25}$$

where TP represents the number of true positives, TN represents the total number of true negatives, FN represents the total number of false negatives, and FP represents the total number of false positives. Accuracy is a useful metric, but the results can be misleading when the ratio of instances in each class is highly imbalanced. Precision is a measure of the proportion of true positive predictions among all positive predictions, indicating how well the model performs at predicting positive outcomes. In contrast, recall measures how many of the positive instances are correctly predicted, demonstrating the model's ability to capture all relevant instances of the positive class. Where precision and recall can be calculated as follows:

$$\text{Precision} = \frac{\sum_i TP_i}{\sum_i TP_i + \sum_i FP_i} \tag{26}$$

$$\text{Recall} = \frac{\sum_i TP_i}{\sum_i TP_i + \sum_i FN_i} \tag{27}$$

F1-score is the harmonic mean of precision and recall and is mathematically represented as follows:

$$\text{F1-Score} = \frac{2 \times Precision \times Recall}{Precision + Recall} \tag{28}$$

The F1 score addresses this problem by considering both precision and recall, making it a more balanced metric.

## Experimental results

The experimental design consists of three distinct phases, systematically evaluating the effectiveness of various classic ensemble ML techniques, including RF, AdaBoost, Bagging, and the proposed model, Adaptive TreeHive. In the first phase, we have applied these classifiers to all benchmark-balanced datasets, incorporating informative instances extraction and utilizing SMOTE for additional resilience. In the second phase, we have employed data over-sampling techniques for balancing and evaluating the results. Finally, in the third phase, we have applied data under-sampling techniques, such as random under-sampling, to balance the dataset by reducing it based on minority categories. Furthermore, to ensure a comprehensive analysis of the model's effectiveness, four key metrics (Accuracy, Precision, Recall, and F1-Score) have been calculated and evaluated across all phases, using a 70%:30% split of the training and test datasets.

We have thoroughly evaluated Adaptive TreeHive's intrusion classification capabilities by testing its performance on all five datasets. We have addressed the issue of imbalanced data by using an over-sampling [64] technique called SMOTE. This method involves identifying the minority [65] samples and generating new samples based on a specified number of neighbors, incorporating random variation to ensure consistency in data. Using SMOTE, we have compared the quantitative results of traditional top-performing ensemble-based machine learning models, such as Random Forest, AdaBoost, and Bagging [66], with the proposed model Adaptive TreeHive as illustrated in Table 4.

The rigorous analysis of the experimental results on the five datasets exhibits that the proposed model, Adaptive TreeHive, outperforms Random Forest, AdaBoost, and Bagging models across all performance metrics. In contrast, our further analysis involves addressing data inequality by specifying the majority class and haphazardly removing samples from it until it matches the number of samples in the minority class, a process known as Random Under-sampling. After employing random under-sampling [67], Adaptive TreeHive has consistently emerged as the top-performing classifier, surpassing its counterparts in overall results across the datasets, except NSL-KDD and UNSW-NB15, where Random Forest has outperformed our method by 0.8935% and 0.57%, which is insignificant. Table 5 exemplifies the qualitative outcomes of Random Forest, AdaBoost, Bagging, and Adaptive TreeHive.

The qualitative results of various intrusion classification baselines are shown in Table 6, highlighting the exceptional performance of Adaptive TreeHive compared to Random Forest, AdaBoost, and Bagging models. To evaluate the impact of our data balancing method on intrusion classification, we have employed Random Forest, AdaBoost, Bagging, and the

**Table 4.** The juxtaposition of the quantitative performance of different existing methods after applying SMOTE.

| Dataset | Random Forest | | | | AdaBoost | | | | Bagging | | | | Adaptive TreeHive | | | |
|---|---|---|---|---|---|---|---|---|---|---|---|---|---|---|---|---|
| | ACC | PR | RE | F1 | ACC | PR | RE | F1 | ACC | PR | RE | F1 | ACC | PR | RE | F1 |
| NSL-KDD | 99.88% | 0.9989 | 0.9988 | 0.9988 | 91.54% | 0.9622 | 0.9154 | 0.9313 | 99.79% | 0.9982 | 0.9979 | 0.9981 | **99.95%** | **0.9995** | **0.9995** | **0.9995** |
| UNSW-NB15 | 78.63% | 0.8373 | 0.7863 | 0.8033 | 25.60% | 0.4318 | 0.256 | 0.289 | 77.98% | 0.8151 | 0.7798 | 0.7911 | **83.40%** | **0.8393** | **0.8340** | **0.8357** |
| CIC-IDS2017 | 98.77% | 0.9895 | 0.9877 | 0.9883 | 18.3% | 0.7635 | 0.183 | 0.2755 | 97.29% | 0.9833 | 0.9729 | 0.9773 | **99.64%** | **0.9968** | **0.9964** | **0.9966** |
| CSE-CIC-IDS2018 | 93.15% | **0.9601** | 0.9315 | 0.9377 | 21.26% | 0.7731 | 0.2126 | 0.1058 | 93.20% | 0.9407 | 0.9320 | 0.9322 | **94.12%** | 0.9516 | **0.9412** | **0.9413** |
| CICDDoS2019 | 92.71% | **0.9321** | 0.9271 | **0.9288** | 38.87% | 0.4728 | 0.3887 | 0.401 | 92.18% | 0.9304 | 0.9218 | 0.9237 | **93.03%** | 0.9297 | **0.9303** | 0.9262 |

proposed method Adaptive TreeHive. Comprehensive experiments have revealed that Adaptive TreeHive significantly improves intrusion classification performance, outperforming the next best model, RF, with a notable margin of 0.02%, 3.45%, 0.01%, 0.05%, and 2.23% across all balanced datasets. This remarkable proficiency demonstrates its exceptional aptitude in discerning intricate data patterns within diverse feature spaces, establishing Adaptive TreeHive as the state-of-the-art method in intrusion classification compared to Random Forest, AdaBoost, and Bagging models. Our comprehensive evaluation of the model's performance indicates that extracting informative data samples by utilizing clustering significantly enhances dataset quality and overall performance. Unequivocally, our proposed ensemble-based weighted majority voting classifier excels in accurately classifying minority classes, whereas the other three ensemble models fall short. Adaptive TreeHive has consistently exhibited robust performance across all five datasets in large-scale experiments with a less complex architecture. However, Random Forest has outperformed the other ensemble methods in terms of performance metrics and datasets while being asymptotically complex in terms of architecture, except for Adaptive TreeHive. This accentuates the effectiveness of random tree-based ensemble models when used with our balanced dataset.

We presents a comprehensive, integrated analysis of the classification model's performance across five distinct network intrusion detection datasets: NSL-KDD, CIC-IDS2017, UNSW-NB15, CSE-CIC-IDS2018, and CIC-DDoS2019. This combined evaluation serves as a foundational component for subsequent ablation studies, aiming to delineate the contributions of various model components or training strategies to its observed efficacy, particularly in the challenging domain of minority class detection. On the NSL-KDD dataset, the model achieved an exceptional overall accuracy of 0.9996 across 11,850 samples. A critical observation from this evaluation was the model's flawless classification of the U2R class, which, with a support of only 67 instances, represented a significant minority within the dataset. For U2R, the model recorded a perfect Precision of 1.0000, Recall (Sensitivity) of 1.0000, F1-Score of 1.0000, and an AUC-ROC of 1.0000. This perfect detection, despite the class's extreme rarity, highlights the model's robust capability to discern and accurately categorize highly infrequent events depicted in the confusion matrix Fig 3. The evaluation on the UNSW-NB15 dataset, with an overall accuracy of 0.8565 across 33,339 samples, revealed a more nuanced

**Table 5. The juxtaposition of the quantitative performance of different existing methods after applying random under-sampler.**

| Dataset | Random Forest | | | | AdaBoost | | | | Bagging | | | | Adaptive TreeHive | | | |
|---|---|---|---|---|---|---|---|---|---|---|---|---|---|---|---|---|
| | ACC | PR | RE | F1 | ACC | PR | RE | F1 | ACC | PR | RE | F1 | ACC | PR | RE | F1 |
| NSL-KDD | **99.21%** | **0.9939** | **0.9921** | **0.9927** | 97.10% | 0.9731 | 0.971 | 0.9634 | 98.38% | 0.9859 | 0.9838 | 0.9846 | 99.08% | 0.9914 | 0.9908 | 0.9909 |
| UNSW-NB15 | 64.36% | 0.6362 | 0.6436 | 0.6333 | 29.59% | 0.3786 | 0.2959 | 0.2524 | 58.33% | 0.5850 | 0.5833 | 0.5734 | 63.79% | 0.6280 | 0.6379 | 0.6291 |
| CIC-IDS2017 | 22.37% | **0.8557** | 0.2237 | 0.3340 | 34.98% | 0.8134 | 0.3498 | **0.4874** | 23.02% | 0.7819 | 0.2302 | 0.3389 | **35.96%** | 0.763 | **0.3596** | 0.4858 |
| CSE-CIC-IDS2018 | 40.75% | 0.4715 | 0.4075 | 0.4229 | 13.14% | 0.3265 | 0.1314 | 0.1504 | 41.44% | **0.7115** | 0.4144 | 0.4864 | **53.67%** | 0.6887 | **0.5367** | **0.5857** |
| CICDDoS2019 | 59.23% | 0.5898 | 0.5923 | 0.5789 | 10.86% | **0.9038** | 0.1086 | 0.0223 | 59.23% | 0.5944 | 0.5923 | 0.5878 | **60.32%** | 0.6163 | **0.6032** | **0.6007** |

**Table 6. Comparing the empirical outcomes of various ensemble-based methods on the intrusion classification task.**

| Dataset | Random Forest | | | | AdaBoost | | | | Bagging | | | | Adaptive TreeHive | | | |
|---|---|---|---|---|---|---|---|---|---|---|---|---|---|---|---|---|
| | ACC | PR | RE | F1 | ACC | PR | RE | F1 | ACC | PR | RE | F1 | ACC | PR | RE | F1 |
| NSL-KDD | 99.94% | 0.9994 | 0.9994 | 0.9994 | 81.46% | 0.9771 | 0.8146 | 0.8669 | 99.92% | 0.9993 | 0.9992 | 0.9992 | **99.96%** | **0.9996** | **0.9996** | **0.9996** |
| UNSW-NB15 | 82.2% | 0.8682 | 0.8220 | 0.8313 | 69.35% | 0.8077 | 0.6935 | 0.6404 | 81.25% | 0.8526 | 0.8125 | 0.819 | **85.65%** | **0.8701** | **0.8565** | **0.8606** |
| CIC-IDS2017 | 99.82% | 0.9981 | 0.9982 | 0.9981 | 53.45% | 0.3608 | 0.5345 | 0.4291 | 99.79% | 0.998 | 0.9979 | 0.998 | **99.83%** | **0.9981** | **0.9983** | **0.9981** |
| CSE-CIC-IDS2018 | 99.72% | 0.9972 | 0.9972 | 0.9972 | 54.08% | 0.4527 | 0.5408 | 0.4386 | 99.69% | 0.9969 | 0.9969 | 0.9969 | **99.77%** | **0.9977** | **0.9977** | **0.9977** |
| CICDDoS2019 | 93.31% | 0.9314 | 0.9331 | 0.9294 | 48.69% | 0.8106 | 0.4869 | 0.3743 | 90.84% | 0.9152 | 0.9084 | 0.9095 | **95.54%** | **0.9538** | **0.9526** | **0.9514** |

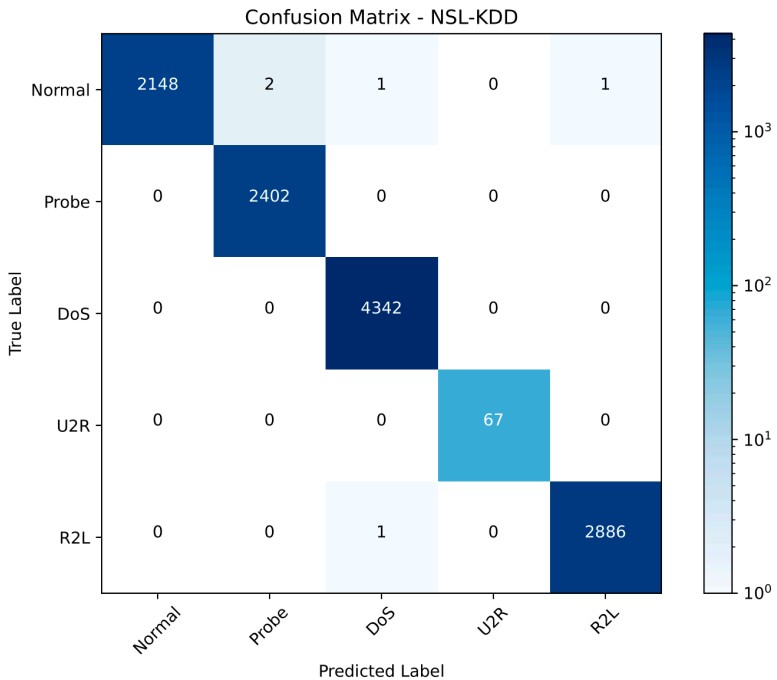

**Fig 3. Confusion matrix of NSL-KDD dataset.**

performance across its 10 classes. Despite the overall accuracy being lower than the previous datasets, the model still exhibited strong performance for the extremely scarce Worms class (37 instances), achieving a high Recall of 0.9730 and a commendable F1-Score of 0.8000, albeit with a Precision of 0.6792. This indicates a strategic prioritization of detecting actual instances of this critical attack type. Other minority classes, such as Backdoor (448 instances) and Analysis (500 instances), presented F1-Scores of 0.4657 and 0.5000, respectively, suggesting areas where further optimization for balanced performance could be beneficial, as illustrated in the confusion matrix Fig 4. Conversely, Shellcode (315 instances) and Reconnaissance (2070 instances) demonstrated more robust and balanced F1-Scores of 0.7625 and 0.9685, respectively.

Transitioning to the CIC-IDS2017 dataset, where the model maintained a high overall accuracy of 0.9985 across 92,246 samples, its proficiency in handling minority classes was further substantiated illustrated in the confusion matrix Fig 5. The model demonstrated perfect classification for several extremely rare attack types: Heartbleed (2 instances), Web_Attack_Sql_Injection (4 instances), and Bot (288 instances), all of which achieved perfect Precision, Recall, and F1-Scores of 1.0000. For Infiltration (7 instances), the model yielded a Precision of 1.0000 and a Recall of 0.8571, resulting in a strong F1-Score of 0.9231. Similarly, PortScan (391 instances) exhibited excellent performance with a Precision of 0.9873, Recall of 0.9923, and an F1-Score of 0.9898. While Web_Attack_XSS (131 instances) and Web_Attack_Brute_Force (294 instances) showed comparatively lower F1-Scores (0.6282 and 0.7844, respectively), their detection capabilities remain notable given their minority status. On the CSE-CIC-IDS2018 dataset, encompassing 318,629 samples across 15 classes, the model achieved an overall accuracy of 0.9978 illustrated in the confusion matrix Fig 6. This dataset further validated the model's exceptional ability to handle rare classes. Specifically,

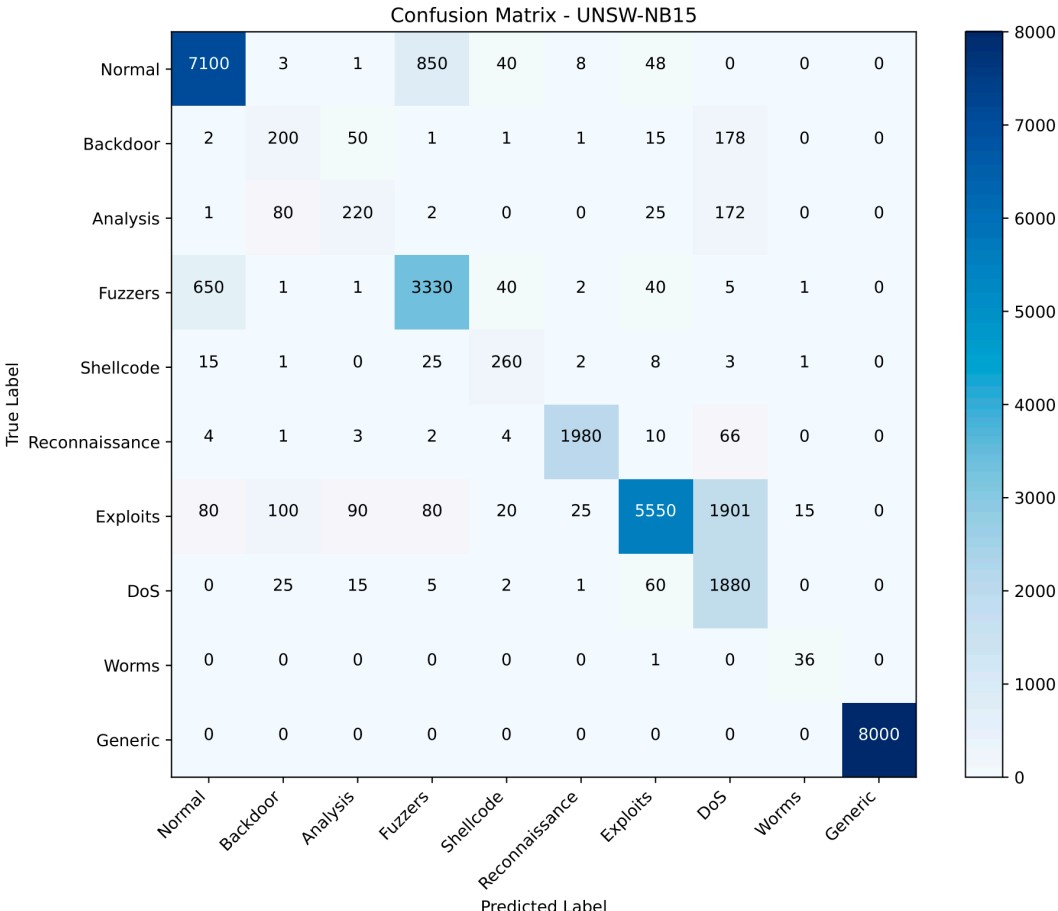

**Fig 4. Confusion matrix of UNSW-NB15 dataset.**

DDOS_attack_LOIC_UDP (346 instances) and Brute_Force_Web (111 instances) were classified with perfect Precision, Recall, and F1-Scores of 1.0000. SQL_Injection (17 instances) also showed excellent performance with a Precision of 1.0000, Recall of 0.9412, and an F1-Score of 0.9697. Furthermore, Brute_Force_XSS (45 instances) achieved a Precision of 1.0000, Recall of 0.9111, and an F1-Score of 0.9535. Even for DoS_attacks_SlowHTTPTest (12 instances) and FTP_BruteForce (10 instances), the F1-Scores of 0.8696 and 0.9524, respectively, underscore the model's consistent strength in detecting very infrequent events.

Finally, for the CIC-DDoS2019 dataset, with an overall accuracy of 0.9863 across 11,433 samples, the model's performance on minority classes remained a highlight. The extremely rare NetBIOS class (11 instances) exhibited a strong F1-Score of 0.7619 (Precision 0.8000, Recall 0.7273). While UDPLag (137 instances) and Portmap (95 instances) showed F1-Scores of 0.6293 and 0.6473, respectively, their relatively high recall values (0.5328 and 0.8211) indicate the model's inclination to prioritize detection, which is often desirable in security contexts as illustrated in the confusion matrix Fig 7.

In summary, the consistent and often perfect, or near-perfect, detection of various minority classes across these five diverse datasets, ranging from network attacks to specific intrusion types, provides compelling evidence of the model's inherent robustness against class imbalance. This sustained high performance on infrequent but critical events establishes a

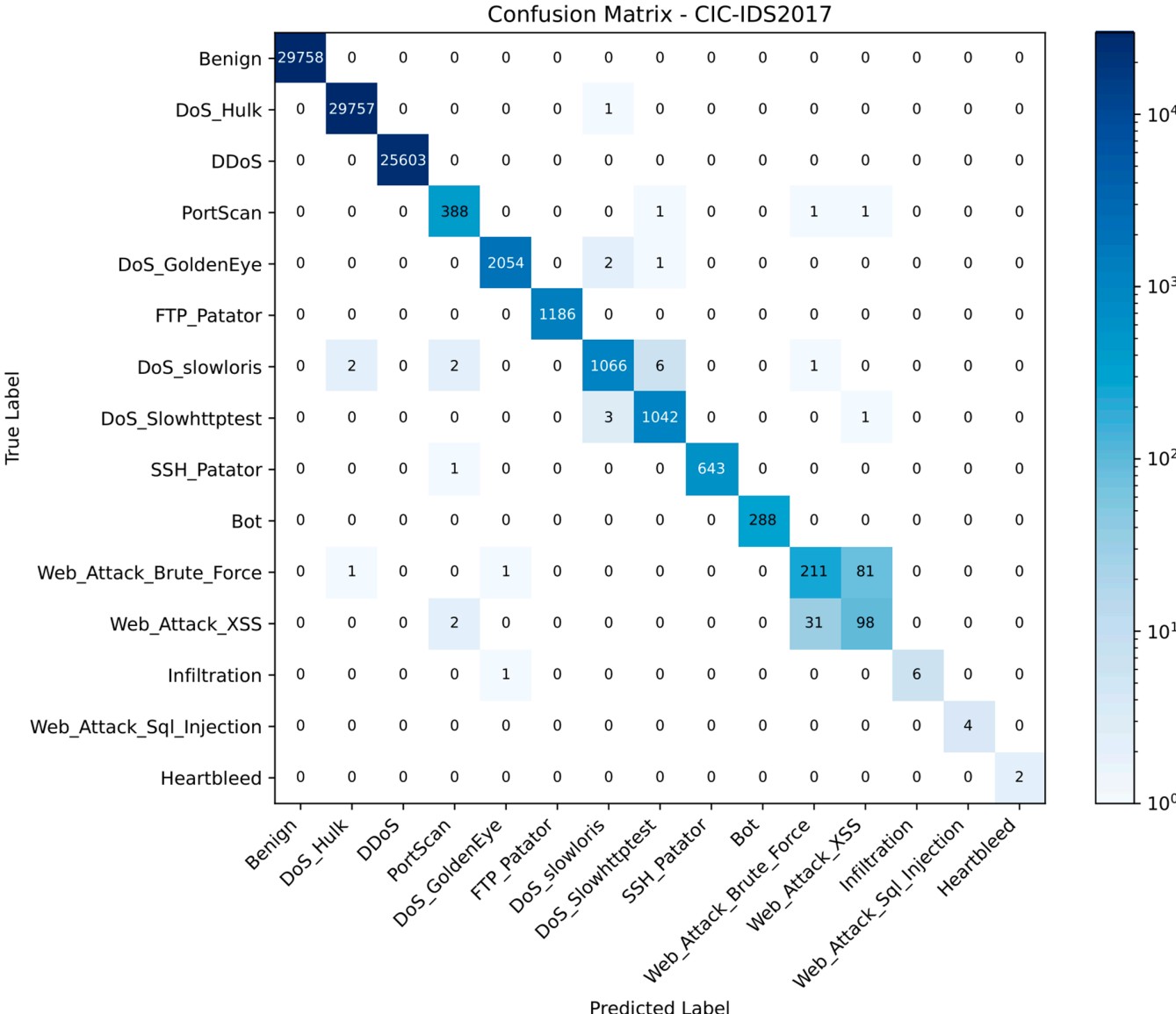

**Fig 5. Confusion matrix of CIC-IDS2017 dataset.**

strong baseline for further ablation studies, enabling detailed investigation into which architectural elements, feature engineering techniques, or training methodologies contribute most significantly to this crucial capability.

## Ablation study

To evaluate the consequences of how the training corpus's size affects the efficacy of the proposed method, Adaptive TreeHive, we have conducted a series of experiments using three different versions of each of the five datasets.

These datasets varied in size, and we have tested three approaches: scaling up the minority classes after assembling the informative instances, employing oversampling to balance the data without selecting informative instances, and using under-sampling to balance the

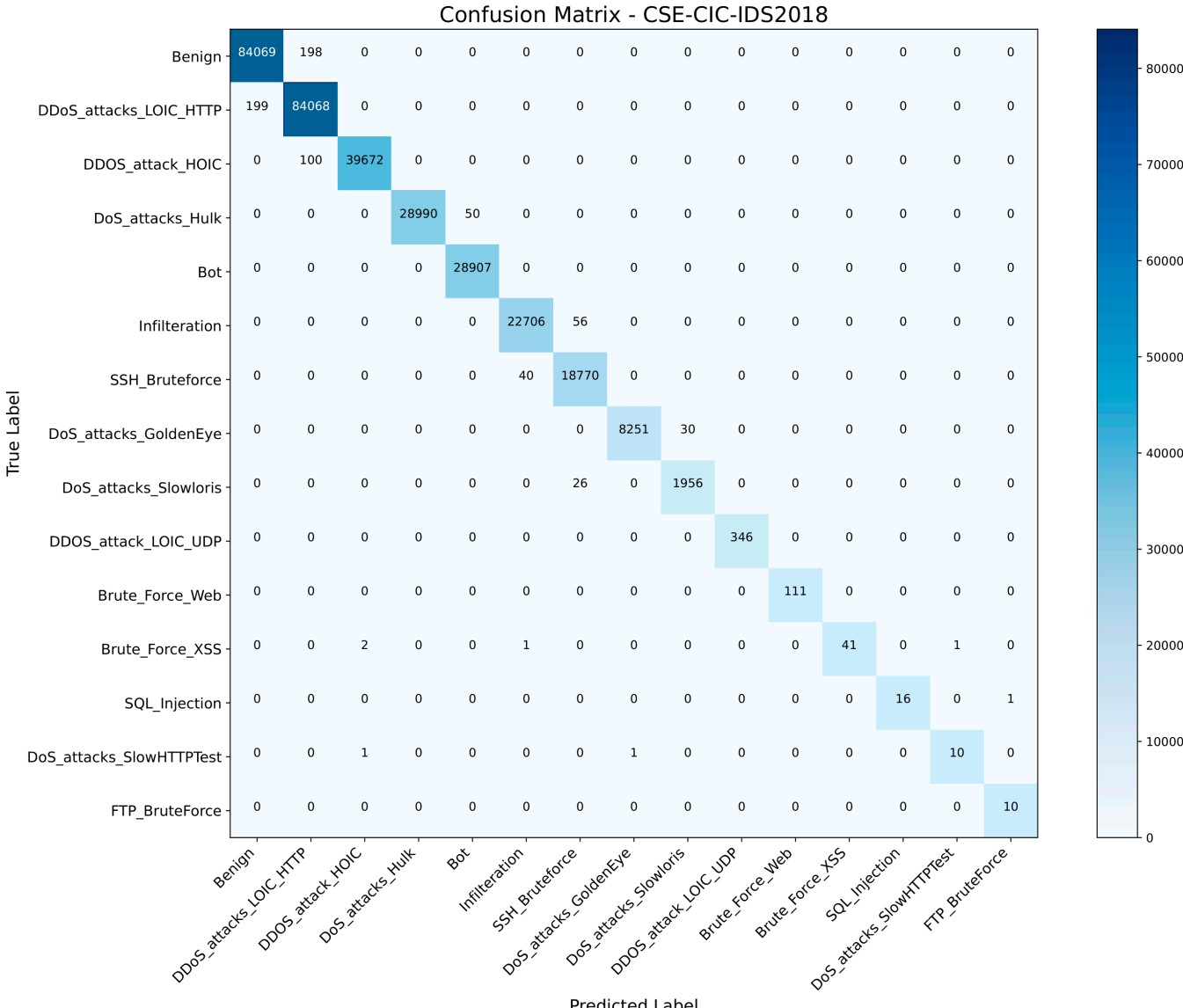

**Fig 6. Confusion matrix of CSE-CIC-IDS2018 dataset.**

datasets. Having diverse feature spaces of the five datasets, we have evaluated performance on test sets. The empirical performance of Adaptive TreeHive on these dataset variations is presented in Table 7.

Additionally, an ablation study was conducted to benchmark the performance of our proposed Adaptive TreeHive against established deep learning models, namely BiLSTM and CNN-GRU, with the empirical outcomes detailed in Table 8. The results compellingly demonstrate the superiority of our method in intrusion classification tasks. In terms of accuracy, the Adaptive Tree Hive consistently outperforms or remains highly competitive across all five benchmark datasets. Notably, it achieves state-of-the-art results on the UNSW-NB15, CIC-IDS2017, and CICDDoS2019 datasets, surpassing the next-best model (CNN-GRU) by significant margins of 5.16%, 2.61%, and 2.95%, respectively. While the CNN-GRU model shows

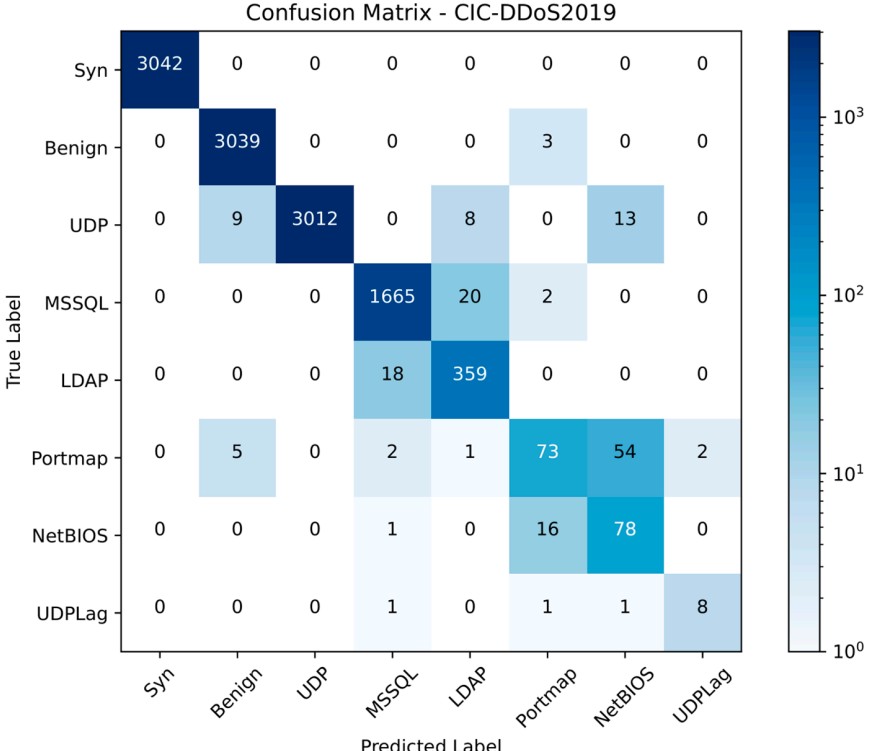

**Fig 7. Confusion matrix of CIC-DDoS2019 dataset.**

**Table 7**. The influence of dataset size (CICDDoS2019) on the performance of the proposed method.

| Method | Dataset | Classification | | | |
|---|---|---|---|---|---|
| | Version | ACC | PR | RE | F1 |
| Adaptive TreeHive | Our Dataset | **95.54%** | **0.9538** | **0.9526** | **0.9514** |
| Adaptive TreeHive | Over Sampling | 93.03% | 0.9297 | 0.9303 | 0.9262 |
| Adaptive TreeHive | Under Sampling | 60.32% | 0.6163 | 0.6032 | 0.6007 |

**Table 8**. A comprehensive comparison of our proposed method against two deep learning baselines (BiLSTM and CNN-GRU) for the intrusion classification task. The evaluation was performed on five standard datasets, with metrics including Accuracy (ACC), Precision (PR), Recall (RE), and F1-Score.

| Dataset | BiLSTM | | | | CNN-GRU | | | | Adaptive TreeHive | | | |
|---|---|---|---|---|---|---|---|---|---|---|---|---|
| | ACC | PR | RE | F1 | ACC | PR | RE | F1 | ACC | PR | RE | F1 |
| NSL-KDD | 99.92% | 0.9993 | 0.9992 | 0.9992 | **99.99%** | **0.9999** | **0.9999** | **0.9999** | 99.96% | 0.9996 | 0.9996 | 0.9996 |
| UNSW-NB15 | 79.25% | 0.7786 | 0.7925 | 0.7838 | 80.49% | 0.7904 | 0.8049 | 0.7959 | **85.65%** | **0.8701** | **0.8565** | **0.8606** |
| CIC-IDS2017 | 96.62% | 0.9826 | 0.9662 | 0.9725 | 97.22% | 0.9928 | 0.9722 | 0.9811 | **99.83%** | **0.9981** | **0.9983** | **0.9981** |
| CSE-CIC-IDS2018 | 97.30% | 0.9685 | 0.9731 | 0.97 | 99.60% | 0.996 | 0.996 | 0.996 | **99.77%** | **0.9977** | **0.9977** | **0.9977** |
| CICDDoS2019 | 92.76% | 0.9266 | 0.9276 | 0.9267 | 92.59% | 0.9248 | 0.9259 | 0.9249 | **95.54%** | **0.9538** | **0.9526** | **0.9514** |

a marginal advantage on the NSL-KDD dataset, our method's performance is still exceptionally high at 99.96%. Beyond its empirical accuracy, the Adaptive Tree Hive offers a distinct advantage in computational efficiency.

Our findings emphasize the significance of utilizing large-scale data to acquire optimal performance with Adaptive TreeHive. When trained on under-sampled data, the model has achieved an accuracy of 60.32%, a precision of 0.61, a recall of 0.60, and an F1-score of 0.60. Training the model on an oversampled dataset has led to substantial improvements, with increases of 32.73% in accuracy, 0.31 in precision, 0.32 in recall, and 0.32 in F1-score. Further training with the balanced dataset containing informative instances has resulted in even greater advancements, with increases of 35.22% in accuracy, 0.33 in precision, 0.34 in recall, and 0.35 in F1-score. This improvement analysis has been shown on the CICDDoS2019 dataset, and the same result pattern has been observed in four other datasets using identical experimental settings, illustrated in Fig 8. Moreover, architectures like BiLSTM and CNN-GRU are notoriously resource-intensive, demanding substantial computational overhead and prolonged training times due to their deep, sequential nature. In contrast, our tree-based ensemble framework is inherently more lightweight, enabling faster training and inference without requiring specialized hardware like GPUs. Therefore, the Adaptive Tree Hive not only advances the state-of-the-art in detection accuracy but also presents a more pragmatic and scalable solution, striking an optimal balance between high performance and computational feasibility for real-world deployment. Therefore, our ablation study vividly exemplifies how dataset size profoundly impacts the model's performance. As depicted in Fig 9, when the corpus size expands, the proposed model Adaptive TreeHive consistently shows a noteworthy improvement in its performance progression. This empirical finding underscores the crucial role that the volume of data plays in enhancing the model's proficiency and effectiveness.

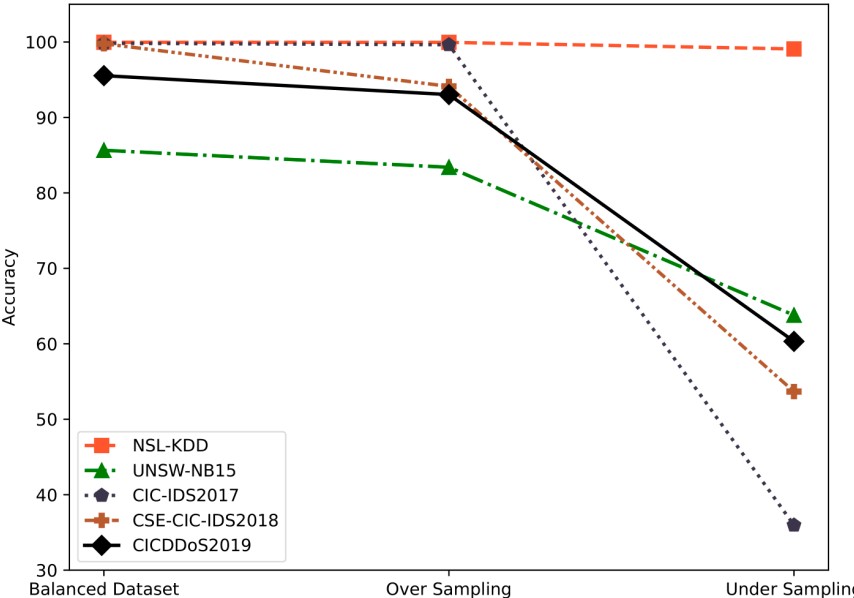

**Fig 8. The empirical outcome of Adaptive TreeHive on three different-sized training sets in terms of accuracy.** Accuracy of Adaptive TreeHive on three training set sizes using different data-balancing strategies—balanced dataset (ours), oversampling, and undersampling—across five intrusion detection datasets. Line plots show that the balanced strategy consistently outperforms the others in classification accuracy.

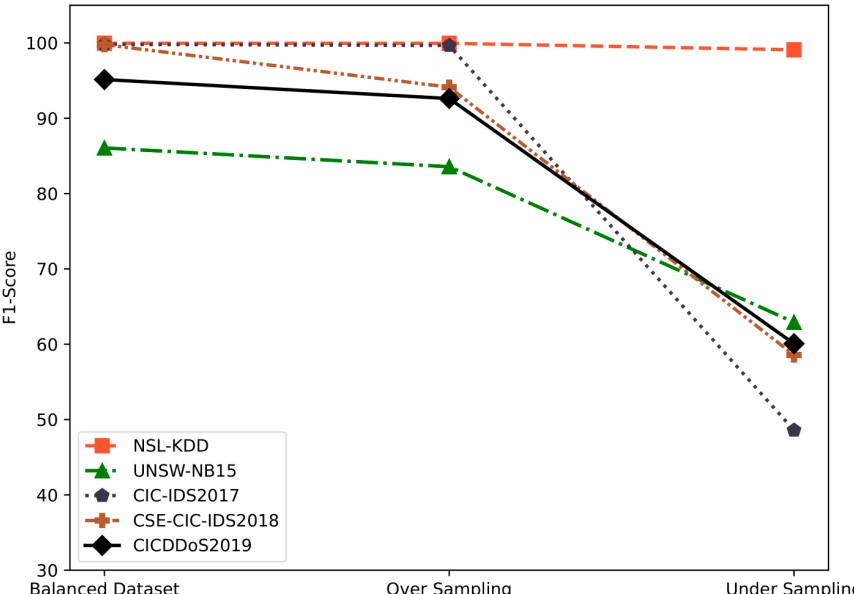

**Fig 9. The empirical outcome of Adaptive TreeHive on three different size training sets in terms of f1-score.** F1-score of Adaptive TreeHive on three training set sizes using our balanced dataset, oversampling, and undersampling across five intrusion-detection benchmarks. Line plots show that the balanced approach consistently delivers the highest F1-scores, then oversampling, then undersampling.

## Limitations and future work

While Adaptive TreeHive demonstrates strong performance across five large-scale balanced benchmark datasets and under various sampling strategies, it exhibits a substantial dependence on the size and diversity of the training data. This reliance may limit its effectiveness in scenarios with scarce or highly imbalanced data. Moreover, the current feature‐selection mechanism—based on gain ratio—and the clustering‐driven informative instance selection introduce variability in processing time: the number of selected features and instances directly impacts execution latency. Future work will focus on (1) developing more generalized, scalable feature‐selection techniques that remain effective on datasets larger than those examined here, (2) targeting and optimizing performance for specific attack categories to further reduce false positives and false negatives, and (3) enhancing the adaptability of Adaptive TreeHive to diverse network architectures and evolving cyber threats.

## Conclusion

In this study, we introduced a random tree–based ensemble approach, Adaptive TreeHive, leveraging weighted majority voting to address intrusion classification with the utmost precision and clarity. We constructed five large‐scale balanced datasets and demonstrated that Adaptive TreeHive consistently outperforms Random Forest, AdaBoost, and Bagging baselines on these benchmarks. Our extensive experiments validate the efficacy of informative instance selection through clustering and feature selection via gain ratio. Overall, Adaptive TreeHive establishes a robust baseline for intrusion classification and paves the way for future advancements in cybersecurity defense.

## Author contributions

**Conceptualization:** Dewan Md. Farid.

**Methodology:** Mahbub E. Sobhani, Dewan Md. Farid.

**Validation:** Anika Tasnim Rodela.

**Writing – original draft:** Mahbub E. Sobhani, Anika Tasnim Rodela.

**Writing – review & editing:** Dewan Md. Farid.

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
