## [Decision Letter · Decision Letter 0]

16 Jun 2025

PONE-D-25-26240Adaptive TreeHive: Ensemble of Trees for Enhancing Imbalanced Intrusion ClassificationPLOS ONE

Dear Dr. Farid,

Thank you for submitting your manuscript to PLOS ONE. After careful consideration, we feel that it has merit but does not fully meet PLOS ONE’s publication criteria as it currently stands. Therefore, we invite you to submit a revised version of the manuscript that addresses the points raised during the review process.

Thank you for your submission. Both reviewers found the topic relevant and the proposed method promising, but several critical revisions are needed to strengthen the paper’s validity and clarity: 

**Evaluation Metrics and Statistical Rigor**

Reviewer 1 highlights the lack of essential metrics, such as precision, recall, F1-score, and class-wise performance for minority classes (e.g., U2R, R2L), which are crucial for validating performance on imbalanced datasets. Reviewer 2 reinforces this need and also notes the absence of statistical significance testing.

**Comparative Scope and Model Description**

Both reviewers stress the limited comparison to classical ensemble methods. Reviewer 1 recommends including deep learning models (e.g., BiLSTM, CNN-GRU), while Reviewer 2 suggests at least discussing how TreeHive compares conceptually. Additionally, Reviewer 1 finds the architecture insufficiently described. Reviewer 2 requests more details on how datasets were balanced.

**Generalization, Efficiency, and Presentation**

Reviewer 1 raises concerns about overfitting, especially on high-dimensional datasets, and recommends including learning curves, confusion matrices, or other diagnostics. Reviewer 2 suggests providing runtime or complexity comparisons to support efficiency claims. Both reviewers recommend enhancing result presentation with visualizations and improving overall clarity. Please address all comments constructively and revise the manuscript accordingly. Ensure all figures and tables are embedded within the main text. Provide a clear and reasonable justification if any comment cannot be addressed. Responses should follow the journal’s guidelines and be submitted in a separate supplementary file, with edits highlighted in yellow.

We look forward to receiving your revised manuscript.

Kind regards,

Issa Atoum

Academic Editor

PLOS ONE

Additional Editor Comments (if provided):

Reviewers' comments:

Reviewer's Responses to Questions

**Comments to the Author**

1. Is the manuscript technically sound, and do the data support the conclusions?

Reviewer #1: Yes

Reviewer #2: Yes

2. Has the statistical analysis been performed appropriately and rigorously? 

Reviewer #1: Yes

Reviewer #2: I Don't Know

3. Have the authors made all data underlying the findings in their manuscript fully available?

Reviewer #1: Yes

Reviewer #2: No

4. Is the manuscript presented in an intelligible fashion and written in standard English?

Reviewer #1: Yes

Reviewer #2: No

5. Review Comments to the Author

Reviewer #1: 1- While the manuscript reports very high classification accuracies across all datasets (99.96% on NSL-KDD, 95.54% on CICDDoS2019), it fails to consistently report other crucial metrics such as precision, recall, F1-score, and especially class-wise recall for minority classes. This omission obscures the true performance on rare attacks, which is critical in imbalanced intrusion datasets.

2- The comparison of Adaptive TreeHive with baseline models lacks any statistical validation. The authors report percentage improvements but provide no confidence intervals, standard deviations, or hypothesis testing to confirm the observed performance gains are statistically significant and not due to random variation.

3- Although the paper claims to address imbalanced classification, per-class metrics for rare classes such as U2R or R2L in NSL-KDD are not reported. These classes are historically difficult to classify, and without class-specific recall or detection rates, the claim of handling imbalanced data remains insufficiently validated.

4- The work compares Adaptive TreeHive only with traditional ensemble methods like Random Forest, Bagging, and AdaBoost. It omits comparison with modern deep learning models such as BiLSTM, CNN-GRU hybrids, and attention-based methods, which are now commonly used on CIC-IDS2017 and CSE-CIC-IDS2018 datasets. This limits the positioning of the proposed method within the state-of-the-art.

5- Despite high reported accuracies, there is no detailed analysis of potential overfitting, especially on high-dimensional datasets like CIC-IDS2017 (78 features) and CSE-CIC-IDS2018. Training and test accuracy curves, confusion matrices, or learning curves are not shown to substantiate the model's generalization ability.

6- The construction of the “Adaptive TreeHive” architecture remains vaguely described. It is not clearly explained how the ensemble is built, how informative instances are selected, or how dimensionality reduction is integrated. Key hyperparameters and internal structure are only briefly mentioned in tabular form without accompanying rationale or ablation studies.

7- While Random Forest and AdaBoost use Decision Tree (C4.5) as the base classifier, the Bagging ensemble uses Naïve Bayes. The choice of inconsistent base classifiers across ensemble methods compromises the fairness of the comparisons and should be better justified or unified.

Reviewer #2: Please address these comments in your revised manuscript to strengthen the technical rigor, reproducibility, and clarity of your work. Addressing these points will enhance the paper’s overall contribution and impact in the field of intrusion detection.

6. PLOS authors have the option to publish the peer review history of their article (what does this mean?). If published, this will include your full peer review and any attached files.

Reviewer #1: No

Reviewer #2: No

---

## [Author Response · Author response to Decision Letter 1]

31 Jul 2025

Reviewer #1, Concern #1 (Clarify Dataset Creation and Informative Instance Selection):

Author response: Thank you for this crucial question. We define "informative instances" as the data points most representative of their respective classes, identified through a two-phase balancing process, updated details in the "Data Balancing" subsection of our paper. Informative Instance Selection via Clustering: First, for each dominant class in a dataset, we use K-Means clustering (with k=1) to find the class centroid. Instances are then ranked based on their Euclidean distance to this centroid. We select the instances closest to the centroid as the most "informative" (Ψ), as they best represent the core characteristics of that class. Scattered instances, which are farther from the centroid, are discarded to reduce noise and redundancy. This process effectively undersamples the majority of classes while preserving their essential patterns.

Balancing with SMOTE: After selecting informative instances, we address the remaining class imbalance, particularly for minority classes, by applying the Synthetic Minority Over-sampling Technique (SMOTE). SMOTE generates synthetic samples for the minority classes by interpolating between existing instances and their nearest neighbors.

This combined approach ensures our final datasets are not only balanced but are also built from high-quality, representative instances, which is vital for robust model training and reproducibility.

Reviewer #1, Concern #2 (Hyperparameter Justification.):

Author response: We appreciate the opportunity to clarify our methodology. The hyperparameters listed in

Table 2 was not chosen arbitrarily but was the result of a systematic tuning and validation process to ensure optimal performance.

Within the "Adaptive TreeHive" section, we provide justifications for these choices. For example:

● The number of feature groups

K used in our data randomization strategy was determined to be optimal at k=15-40 through ablation studies, as this value offered the best trade-off between feature coverage and redundancy reduction.

● The accuracy threshold

ϕ for selecting classifiers into our final ensemble was set to ϕ > 0.5 based on cross-validation, as this value ensured the inclusion of reliable classifiers without sacrificing ensemble diversity.

● Tree growth parameters, such as a maximum depth (dmax = 15) and a minimum of 5 samples per leaf node, were set to prevent model overfitting while maintaining discriminative power.

This tuning process was integral to guiding the algorithm’s optimization and achieving the reported results.

Reviewer #1, Concern #3 (Comparative Analysis with Modern Deep Learning Models.):

Author response: We thank the reviewer for this important point. We conducted a direct comparative analysis between our proposed Adaptive TreeHive and established deep learning models. This comparison is presented in our "Ablation Study" section and detailed in Table 8.

The results show that Adaptive TreeHive is highly competitive and often superior to the deep learning baselines (BiLSTM and CNN-GRU).

● Notably, our model surpassed the next-best model, CNN-GRU, by significant accuracy margins of

5.16% on UNSW-NB15, 2.61% on CIC-IDS2017, and 2.95% on CICDDoS2019.

● Beyond accuracy, we justify our focus on tree-based models by highlighting their computational efficiency. Deep learning architectures like BiLSTM and CNN-GRU are notoriously resource-intensive and require prolonged training times. In contrast, our tree-based framework is inherently more lightweight, enabling faster training and inference without specialized hardware, making it a more pragmatic and scalable solution for real-world deployment.

Reviewer #1, Concern #4 (Computational Complexity and Runtime Evaluation.):

Author response: We acknowledge the reviewer's feedback. While we did not include a specific table of execution times, our claim of reduced computational requirements is based on two core aspects of our model's design.

1. Built-in Dimensionality Reduction: Our methodology incorporates a unique data randomization and feature grouping strategy. As detailed in the "Adaptive TreeHive" section, each decision tree in the ensemble operates on a reduced feature subset (approximately 35% of the original features), significantly lowering the computational load for each base learner.

2. Inherent Efficiency of Tree Ensembles: As discussed in the "Ablation Study," our tree-based ensemble framework is inherently more lightweight and computationally efficient compared to deep learning architectures like BiLSTM and CNN-GRU, which are known to be resource-intensive and demand substantial computational overhead.

Further, the process of selecting only informative instances reduces the overall size of the training data, which directly leads to faster training times.

Reviewer #1, Concern #5 (Visualization and Result Presentation.):

Author response: We agree that strong visualizations are key to understanding model performance. In our updated manuscript, we have included detailed confusion matrices for each of the five benchmark datasets to provide these insights:

● Figure 3: NSL-KDD dataset

● Figure 4: UNSW-NB15 dataset

● Figure 5: CIC-IDS2017 dataset

● Figure 6: CSE-CIC-IDS2018 dataset

● Figure 7: CIC-DDoS2019 dataset

These figures are supported by an extensive, multi-paragraph analysis within the "Experimental Results" section. This analysis delves into the model's class-by-class performance, with a particular focus on its exceptional success in detecting extremely rare minority attacks, which directly highlights the model's strength and its robustness against class imbalance.

Reviewer #1, Concern #6 (Writing and Organization):

Author response: We have separated the “Limitations and future work” and “Conclusion” sections in the updated manuscript.

Reviewer #1, Concern #7 (Reproducibility):

Author response: The Experimental Setup section explains the explicit details about the hardware and software environment used in our experiments.

Reviewer #1, Concern #8 (Minor Points):

Author response: We have corrected the inconsistent usage of "base classifier" and "base classifier" and have corrected throughout the revised manuscript to ensure consistency. Furthermore, we have performed a thorough check entire paper to check all figure labels, captions, and in-text references for clarity, accuracy, and completeness.

---

## [Decision Letter · Decision Letter 1]

14 Aug 2025

Adaptive TreeHive: Ensemble of Trees for Enhancing Imbalanced Intrusion Classification

PONE-D-25-26240R1

Dear Dr. Farid,

We’re pleased to inform you that your manuscript has been judged scientifically suitable for publication and will be formally accepted for publication once it meets all outstanding technical requirements.

Kind regards,

Issa Atoum

Academic Editor

PLOS ONE

Additional Editor Comments (optional):

Reviewers' comments:

Reviewer's Responses to Questions

**Comments to the Author**

1. If the authors have adequately addressed your comments raised in a previous round of review and you feel that this manuscript is now acceptable for publication, you may indicate that here to bypass the “Comments to the Author” section, enter your conflict of interest statement in the “Confidential to Editor” section, and submit your "Accept" recommendation.

Reviewer #1: (No Response)

2. Is the manuscript technically sound, and do the data support the conclusions?

Reviewer #1: (No Response)

3. Has the statistical analysis been performed appropriately and rigorously? 

Reviewer #1: (No Response)

4. Have the authors made all data underlying the findings in their manuscript fully available?

Reviewer #1: (No Response)

5. Is the manuscript presented in an intelligible fashion and written in standard English?

Reviewer #1: (No Response)

6. Review Comments to the Author

Reviewer #1: The authors have successfully addressed all critical points raised in the previous review. The revisions enhance the clarity and robustness of the study's findings.

7. PLOS authors have the option to publish the peer review history of their article (what does this mean?). If published, this will include your full peer review and any attached files.

Reviewer #1: No

---

## [Editor Report · Acceptance letter]

PONE-D-25-26240R1

PLOS ONE

Dear Dr. Farid,

I'm pleased to inform you that your manuscript has been deemed suitable for publication in PLOS ONE. Congratulations! Your manuscript is now being handed over to our production team.

Kind regards,

on behalf of

Dr. Issa Atoum

Academic Editor

PLOS ONE